# Active Learning from Weak and Strong Labelers

**Chicheng Zhang**
UC San Diego
chichengzhang@ucsd.edu

**Kamalika Chaudhuri**
UC San Diego
kamalika@eng.ucsd.edu

## Abstract

An active learner is given a hypothesis class, a large set of unlabeled examples and the ability to interactively query labels to an oracle of a subset of these examples; the goal of the learner is to learn a hypothesis in the class that fits the data well by making as few label queries as possible.

This work addresses active learning with labels obtained from strong and weak labelers, where in addition to the standard active learning setting, we have an extra weak labeler which may occasionally provide incorrect labels. An example is learning to classify medical images where either expensive labels may be obtained from a physician (oracle or strong labeler), or cheaper but occasionally incorrect labels may be obtained from a medical resident (weak labeler). Our goal is to learn a classifier with low error on data labeled by the oracle, while using the weak labeler to reduce the number of label queries made to this labeler. We provide an active learning algorithm for this setting, establish its statistical consistency, and analyze its label complexity to characterize when it can provide label savings over using the strong labeler alone.

## 1   Introduction

An active learner is given a hypothesis class, a large set of unlabeled examples and the ability to interactively make label queries to an oracle on a subset of these examples; the goal of the learner is to learn a hypothesis in the class that fits the data well by making as few oracle queries as possible.

As labeling examples is a tedious task for any one person, many applications of active learning involve synthesizing labels from multiple experts who may have slightly different labeling patterns. While a body of recent empirical work [27, 28, 29, 25, 26, 11] has developed methods for combining labels from multiple experts, little is known on the theory of actively learning with labels from multiple annotators. For example, what kind of assumptions are needed for methods that use labels from multiple sources to work, when these methods are statistically consistent, and when they can yield benefits over plain active learning are all open questions.

This work addresses these questions in the context of active learning from strong and weak labelers. Specifically, in addition to unlabeled data and the usual labeling oracle in standard active learning, we have an extra weak labeler. The labeling oracle is a *gold standard* – an expert on the problem domain – and it provides high quality but expensive labels. The weak labeler is cheap, but may provide incorrect labels on some inputs. An example is learning to classify medical images where either expensive labels may be obtained from a physician (oracle), or cheaper but occasionally incorrect labels may be obtained from a medical resident (weak labeler). Our goal is to learn a classifier in a hypothesis class whose error with respect to the data labeled by the oracle is low, while exploiting the weak labeler to reduce the number of queries made to this oracle. Observe that in our model the weak labeler can be incorrect anywhere, and does not necessarily provide uniformly noisy labels everywhere, as was assumed by some previous works [7, 23].

A plausible approach in this framework is to learn a *difference classifier* to predict where the weak labeler differs from the oracle, and then use a standard active learning algorithm which queries the weak labeler when this difference classifier predicts agreement. Our first key observation is that this approach is *statistically inconsistent*; false negative errors (that predict no difference when $O$ and $W$ differ) lead to biased annotation for the target classification task. We address this problem by learning instead a *cost-sensitive difference classifier* that ensures that false negative errors rarely occur. Our second key observation is that as existing active learning algorithms usually query labels in localized regions of space, it is sufficient to train the difference classifier restricted to this region and still maintain consistency. This process leads to significant label savings. Combining these two ideas, we get an algorithm that is provably statistically consistent and that works under the assumption that there is a good difference classifier with low false negative error.

We analyze the label complexity of our algorithm as measured by the number of label requests to the labeling oracle. In general we cannot expect any consistent algorithm to provide label savings under all circumstances, and indeed our worst case asymptotic label complexity is the same as that of active learning using the oracle alone. Our analysis characterizes when we can achieve label savings, and we show that this happens for example if the weak labeler agrees with the labeling oracle for some fraction of the examples close to the decision boundary. Moreover, when the target classification task is agnostic, the number of labels required to learn the difference classifier is of a lower order than the number of labels required for active learning; thus in realistic cases, learning the difference classifier adds only a small overhead to the total label requirement, and overall we get label savings over using the oracle alone.

**Related Work.** There has been a considerable amount of empirical work on active learning where multiple annotators can provide labels for the unlabeled examples. One line of work assumes a generative model for each annotator's labels. The learning algorithm learns the parameters of the individual labelers, and uses them to decide which labeler to query for each example. [28, 29, 12] consider separate logistic regression models for each annotator, while [18, 19] assume that each annotator's labels are corrupted with a different amount of random classification noise. A second line of work [11, 15] that includes Pro-Active Learning, assumes that each labeler is an expert over an unknown subset of categories, and uses data to measure the class-wise expertise in order to optimally place label queries. In general, it is not known under what conditions these algorithms are statistically consistent, particularly when the modeling assumptions do not strictly hold, and under what conditions they provide label savings over regular active learning.

[24], the first theoretical work to consider this problem, consider a model where the weak labeler is more likely to provide incorrect labels in heterogeneous regions of space where similar examples have different labels. Their formalization is orthogonal to ours – while theirs is more natural in a non-parametric setting, ours is more natural for fitting classifiers in a hypothesis class. In a NIPS 2014 Workshop paper, [20] have also considered learning from strong and weak labelers; unlike ours, their work is in the online selective sampling setting, and applies only to linear classifiers and robust regression. [10] study learning from multiple teachers in the online selective sampling setting in a model where different labelers have different regions of expertise.

Finally, there is a large body of theoretical work [1, 8, 9, 13, 30, 2, 4] on learning a binary classifier based on interactive label queries made *to a single labeler*. In the realizable case, [21, 8] show that a generalization of binary search provides an exponential improvement in label complexity over passive learning. The problem is more challenging, however, in the more realistic agnostic case, where such approaches lead to inconsistency. The two styles of algorithms for agnostic active learning are disagreement-based active learning (DBAL) [1, 9, 13, 4] and the more recent margin-based or confidence-based active learning [2, 30]. Our algorithm builds on recent work in DBAL [4, 14].

## 2    Preliminaries

**The Model.** We begin with a general framework for actively learning from weak and strong labelers. In the standard active learning setting, we are given unlabelled data drawn from a distribution $U$ over an input space $\mathscr{X}$, a label space $\mathscr{Y} = \{-1, 1\}$, a hypothesis class $\mathscr{H}$, and a labeling oracle $O$ to which we can make interactive queries.

In our setting, we additionally have access to a weak labeling oracle $W$ which we can query interactively. Querying $W$ is significantly cheaper than querying $O$; however, querying $W$ generates a label $y_W$ drawn from a conditional distribution $\mathbb{P}_W(y_W|x)$ which is not the same as the conditional distribution $\mathbb{P}_O(y_O|x)$ of $O$.

Let $D$ be the data distribution over labelled examples such that: $\mathbb{P}_D(x,y) = \mathbb{P}_U(x)\mathbb{P}_O(y|x)$. Our goal is to learn a classifier $h$ in the hypothesis class $\mathcal{H}$ such that with probability $\geq 1 - \delta$ over the sample, we have: $\mathbb{P}_D(h(x) \neq y) \leq \min_{h' \in \mathcal{H}} \mathbb{P}_D(h'(x) \neq y) + \varepsilon$, while making as few (interactive) queries to $O$ as possible.

Observe that in this model $W$ may disagree with the oracle $O$ *anywhere* in the input space; this is unlike previous frameworks [7, 23] where labels assigned by the weak labeler are corrupted by random classification noise with a higher variance than the labeling oracle. We believe this feature makes our model more realistic.

Second, unlike [24], mistakes made by the weak labeler *do not have to be close to the decision boundary*. This keeps the model general and simple, and allows greater flexibility to weak labelers. Our analysis shows that if $W$ is largely incorrect close to the decision boundary, then our algorithm will automatically make more queries to $O$ in its later stages.

Finally note that $O$ is allowed to be *non-realizable* with respect to the target hypothesis class $\mathcal{H}$.

**Background on Active Learning Algorithms.** The standard active learning setting is very similar to ours, the only difference being that we have access to the weak oracle $W$. There has been a long line of work on active learning [1, 6, 8, 13, 2, 9, 4, 30]. Our algorithms are based on a style called *disagreement-based active learning (DBAL)*. The main idea is as follows. Based on the examples seen so far, the algorithm maintains a candidate set $V_t$ of classifiers in $\mathcal{H}$ that is guaranteed with high probability to contain $h^*$, the classifier in $\mathcal{H}$ with the lowest error. Given a randomly drawn unlabeled example $x_t$, if all classifiers in $V_t$ agree on its label, then this label is inferred; observe that with high probability, this inferred label is $h^*(x_t)$. Otherwise, $x_t$ is said to be in the *disagreement region* of $V_t$, and the algorithm queries $O$ for its label. $V_t$ is updated based on $x_t$ and its label, and algorithm continues.

Recent works in DBAL [9, 4] have observed that it is possible to determine if an $x_t$ is in the disagreement region of $V_t$ without explicitly maintaining $V_t$. Instead, a labelled dataset $S_t$ is maintained; the labels of the examples in $S_t$ are obtained by either querying the oracle or direct inference. To determine whether an $x_t$ lies in the disagreement region of $V_t$, two constrained ERM procedures are performed; empirical risk is minimized over $S_t$ while constraining the classifier to output the label of $x_t$ as 1 and $-1$ respectively. If these two classifiers have similar training errors, then $x_t$ lies in the disagreement region of $V_t$; otherwise the algorithm infers a label for $x_t$ that agrees with the label assigned by $h^*$.

**More Definitions and Notation.** The error of a classifier $h$ under a labelled data distribution $Q$ is defined as: $\mathrm{err}_Q(h) = \mathbb{P}_{(x,y) \sim Q}(h(x) \neq y)$; we use the notation $\mathrm{err}(h, S)$ to denote its empirical error on a labelled data set $S$. We use the notation $h^*$ to denote the classifier with the lowest error under $D$ and $\nu$ to denote its error $\mathrm{err}_D(h^*)$, where $D$ is the target labelled data distribution.

Our active learning algorithm implicitly maintains a $(1-\delta)$-*confidence set* for $h^*$ throughout the algorithm. Given a set $S$ of labelled examples, a set of classifiers $V(S) \subseteq \mathcal{H}$ is said to be a $(1-\delta)$-confidence set for $h^*$ with respect to $S$ if $h^* \in V$ with probability $\geq 1 - \delta$ over $S$.

The disagreement between two classifiers $h_1$ and $h_2$ under an unlabelled data distribution $U$, denoted by $\rho_U(h_1, h_2)$, is $\mathbb{P}_{x \sim U}(h_1(x) \neq h_2(x))$. Observe that the disagreements under $U$ form a pseudometric over $\mathcal{H}$. We use $\mathrm{B}_U(h, r)$ to denote a ball of radius $r$ centered around $h$ in this metric. The *disagreement region* of a set $V$ of classifiers, denoted by $\mathrm{DIS}(V)$, is the set of all examples $x \in \mathcal{X}$ such that there exist two classifiers $h_1$ and $h_2$ in $V$ for which $h_1(x) \neq h_2(x)$.

## 3   Algorithm

Our main algorithm is a standard single-annotator DBAL algorithm with a major modification: when the DBAL algorithm makes a label query, we use an extra sub-routine to decide whether this query should be made to the oracle or the weak labeler, and make it accordingly. How do we make this

decision? We try to predict if weak labeler differs from the oracle on this example; if so, query the oracle, otherwise, query the weak labeler.

**Key Idea 1: Cost Sensitive Difference Classifier.** How do we predict if the weak labeler differs from the oracle? A plausible approach is to learn a *difference classifier* $h^{df}$ in a hypothesis class $\mathscr{H}^{df}$ to determine if there is a difference. Our first key observation is when the region where $O$ and $W$ differ cannot be perfectly modeled by $\mathscr{H}^{df}$, the resulting active learning algorithm is *statistically inconsistent*. Any false negative errors (that is, incorrectly predicting no difference) made by difference classifier leads to biased annotation for the target classification task, which in turn leads to inconsistency. We address this problem by instead learning a *cost-sensitive difference classifier* and we assume that a classifier with low false negative error exists in $\mathscr{H}^{df}$. While training, we constrain the false negative error of the difference classifier to be low, and minimize the number of predicted positives (or disagreements between $W$ and $O$) subject to this constraint. This ensures that the annotated data used by the active learning algorithm has diminishing bias, thus ensuring consistency.

**Key Idea 2: Localized Difference Classifier Training.** Unfortunately, even with cost-sensitive training, directly learning a difference classifier accurately is expensive. If $d'$ is the VC-dimension of the difference hypothesis class $\mathscr{H}^{df}$, to learn a target classifier to excess error $\varepsilon$, we need a difference classifier with false negative error $O(\varepsilon)$, which, from standard generalization theory, requires $\tilde{O}(d'/\varepsilon)$ labels [5, 22]! Our second key observation is that we can save on labels by training the difference classifier in a localized manner – because the DBAL algorithm that builds the target classifier *only makes label queries* in the disagreement region of the current confidence set for $h^*$. Therefore we train the difference classifier only on this region and still maintain consistency. Additionally this provides label savings because while training the target classifier to excess error $\varepsilon$, we need to train a difference classifier with only $\tilde{O}(d'\phi_k/\varepsilon)$ labels where $\phi_k$ is the probability mass of this disagreement region. The localized training process leads to an additional technical challenge: as the confidence set for $h^*$ is updated, its disagreement region changes. We address this through an epoch-based DBAL algorithm, where the confidence set is updated and a fresh difference classifier is trained in each epoch.

**Main Algorithm.** Our main algorithm (Algorithm 1) combines these two key ideas, and like [4], implicitly maintains the $(1-\delta)$-confidence set for $h^*$ by through a labeled dataset $\hat{S}_k$. In epoch $k$, the target excess error is $\varepsilon_k \approx \frac{1}{2^k}$, and the goal of Algorithm 1 is to generate a labeled dataset $\hat{S}_k$ that implicitly represents a $(1-\delta_k)$-confidence set on $h^*$. Additionally, $\hat{S}_k$ has the property that the empirical risk minimizer over it has excess error $\leq \varepsilon_k$.

A naive way to generate such an $\hat{S}_k$ is by drawing $\tilde{O}(d/\varepsilon_k^2)$ labeled examples, where $d$ is the VC dimension of $\mathscr{H}$. Our goal, however, is to generate $\hat{S}_k$ using a much smaller number of label queries, which is accomplished by Algorithm 5. This is done in two ways. First, like standard DBAL, we infer the label of any $x$ that lies *outside* the disagreement region of the current confidence set for $h^*$. Algorithm 4 identifies whether an $x$ lies in this region. Second, for any $x$ in the disagreement region, we determine whether $O$ and $W$ agree on $x$ using a difference classifier; if there is agreement, we query $W$, else we query $O$. The difference classifier used to determine agreement is retrained in the beginning of each epoch by Algorithm 2, which ensures that the annotation has low bias.

The algorithms use a constrained ERM procedure CONS-LEARN. Given a hypothesis class $H$, a labeled dataset $S$ and a set of constraining examples $C$, CONS-LEARN$_H(C, S)$ returns a classifier in $H$ that minimizes the empirical error on $S$ subject to $h(x_i) = y_i$ for each $(x_i, y_i) \in C$.

**Identifying the Disagreement Region.** Algorithm 4 (deferred to the Appendix) identifies if an unlabeled example $x$ lies in the disagreement region of the current $(1-\delta)$-confidence set for $h^*$; recall that this confidence set is implicitly maintained through $\hat{S}_k$. The identification is based on two ERM queries. Let $\hat{h}$ be the empirical risk minimizer on the current labeled dataset $\hat{S}_{k-1}$, and $\hat{h}'$ be the empirical risk minimizer on $\hat{S}_{k-1}$ under the constraint that $\hat{h}'(x) = -\hat{h}(x)$. If the training errors of $\hat{h}$ and $\hat{h}'$ are very different, then, all classifiers with training error close to that of $\hat{h}$ assign the same label to $x$, and $x$ lies outside the current disagreement region.

**Training the Difference Classifier.** Algorithm 2 trains a difference classifier on a random set of examples which lies in the disagreement region of the current confidence set for $h^*$. The training process is cost-sensitive, and is similar to [16, 17, 5, 22]. A hard bound is imposed on the false-negative error, which translates to a bound on the annotation bias for the target task. The number of positives (i.e., the number of examples where $W$ and $O$ differ) is minimized subject to this constraint; this amounts to (approximately) minimizing the fraction of queries made to $O$.

The number of labeled examples used in training is large enough to ensure false negative error $O(\varepsilon_k/\phi_k)$ over the disagreement region of the current confidence set; here $\phi_k$ is the probability mass of this disagreement region under $U$. This ensures that the overall annotation bias introduced by this procedure in the target task is at most $O(\varepsilon_k)$. As $\phi_k$ is small and typically diminishes with $k$, this requires less labels than training the difference classifier globally which would have required $\tilde{O}(d'/\varepsilon_k)$ queries to $O$.

---

**Algorithm 1** Active Learning Algorithm from Weak and Strong Labelers

---

1: Input: Unlabeled distribution $U$, target excess error $\varepsilon$, confidence $\delta$, labeling oracle $O$, weak oracle $W$, hypothesis class $\mathcal{H}$, hypothesis class for difference classifier $\mathcal{H}^{df}$.
2: Output: Classifier $\hat{h}$ in $\mathcal{H}$.
3: Initialize: initial error $\varepsilon_0 = 1$, confidence $\delta_0 = \delta/4$. Total number of epochs $k_0 = \lceil \log \frac{1}{\varepsilon} \rceil$.
4: Initial number of examples $n_0 = O(\frac{1}{\varepsilon_0^2}(d \ln \frac{1}{\varepsilon_0^2} + \ln \frac{1}{\delta_0}))$.
5: Draw a fresh sample and query $O$ for its labels $\hat{S}_0 = \{(x_1, y_1), \ldots, (x_{n_0}, y_{n_0})\}$. Let $\sigma_0 = \sigma(n_0, \delta_0)$.
6: **for** $k = 1, 2, \ldots, k_0$ **do**
7:     Set target excess error $\varepsilon_k = 2^{-k}$, confidence $\delta_k = \delta/4(k+1)^2$.
8:     *# Train Difference Classifier*
9:     $\hat{h}_k^{df} \leftarrow$ Call Algorithm 2 with inputs unlabeled distribution $U$, oracles $W$ and $O$, target excess error $\varepsilon_k$, confidence $\delta_k/2$, previously labeled dataset $\hat{S}_{k-1}$.
10:     *# Adaptive Active Learning using Difference Classifier*
11:     $\sigma_k, \hat{S}_k \leftarrow$ Call Algorithm 5 with inputs unlabeled distribution $U$, oracles $W$ and $O$, difference classifier $\hat{h}_k^{df}$, target excess error $\varepsilon_k$, confidence $\delta_k/2$, previously labeled dataset $\hat{S}_{k-1}$.
12: **end for**
13: **return** $\hat{h} \leftarrow \text{CONS-LEARN}_{\mathcal{H}}(\emptyset, \hat{S}_{k_0})$.

---

**Adaptive Active Learning using the Difference Classifier.** Finally, Algorithm 5 (deferred to the Appendix) is our main active learning procedure, which generates a labeled dataset $\hat{S}_k$ that is implicitly used to maintain a tighter $(1 - \delta)$-confidence set for $h^*$. Specifically, Algorithm 5 generates a $\hat{S}_k$ such that the set $V_k$ defined as:

$$V_k = \{h : \text{err}(h, \hat{S}_k) - \min_{\hat{h}_k \in \mathcal{H}} \text{err}(\hat{h}_k, \hat{S}_k) \le 3\varepsilon_k/4\}$$

has the property that:

$$\{h : \text{err}_D(h) - \text{err}_D(h^*) \le \varepsilon_k/2\} \subseteq V_k \subseteq \{h : \text{err}_D(h) - \text{err}_D(h^*) \le \varepsilon_k\}$$

This is achieved by labeling, through inference or query, a large enough sample of unlabeled data drawn from $U$. Labels are obtained from three sources - direct inference (if $x$ lies outside the disagreement region as identified by Algorithm 4), querying $O$ (if the difference classifier predicts a difference), and querying $W$. How large should the sample be to reach the target excess error? If $\text{err}_D(h^*) = \nu$, then achieving an excess error of $\varepsilon$ requires $\tilde{O}(d\nu/\varepsilon_k^2)$ samples, where $d$ is the VC dimension of the hypothesis class. As $\nu$ is unknown in advance, we use a doubling procedure in lines 4-14 to iteratively determine the sample size.

**Algorithm 2** Training Algorithm for Difference Classifier

1: Input: Unlabeled distribution $U$, oracles $W$ and $O$, target error $\varepsilon$, hypothesis class $\mathcal{H}^{df}$, confidence $\delta$, previous labeled dataset $\hat{T}$.
2: Output: Difference classifier $\hat{h}^{df}$.
3: Let $\hat{p}$ be an estimate of $\mathbb{P}_{x \sim U}(\mathbf{in\_disagr\_region}(\hat{T}, \frac{3\varepsilon}{2}, x) = 1)$, obtained by calling Algorithm 3(deferred to the Appendix) with failure probability $\delta/3$. [1]
4: Let $U' = \emptyset$, $i = 1$, and

$$m = \frac{64 \cdot 1024\hat{p}}{\varepsilon}\left(d' \ln \frac{512 \cdot 1024\hat{p}}{\varepsilon} + \ln \frac{72}{\delta}\right) \tag{1}$$

5: **repeat**
6:　　Draw an example $x_i$ from $U$.
7:　　**if** $\mathbf{in\_disagr\_region}(\hat{T}, \frac{3\varepsilon}{2}, x_i) = 1$ **then**　*# $x_i$ is inside the disagreement region*
8:　　　　query both $W$ and $O$ for labels to get $y_{i,W}$ and $y_{i,O}$.
9:　　**end if**
10:　　$U' = U' \cup \{(x_i, y_{i,O}, y_{i,W})\}$
11:　　$i = i + 1$
12: **until** $|U'| = m$
13: Learn a classifier $\hat{h}^{df} \in \mathcal{H}^{df}$ based on the following empirical risk minimizer:

$$\hat{h}^{df} = \operatorname{argmin}_{h^{df} \in \mathcal{H}^{df}} \sum_{i=1}^{m} \mathbb{1}(h^{df}(x_i) = +1), \text{ s.t. } \sum_{i=1}^{m} \mathbb{1}(h^{df}(x_i) = -1 \wedge y_{i,O} \neq y_{i,W}) \leq m\varepsilon/256\hat{p} \tag{2}$$

14: **return** $\hat{h}^{df}$.

## 4　Performance Guarantees

We now examine the performance of our algorithm, which is measured by the number of label queries made to the oracle $O$. Additionally we require our algorithm to be statistically consistent, which means that the true error of the output classifier should converge to the true error of the best classifier in $\mathcal{H}$ on the data distribution $D$.

Since our framework is very general, we cannot expect any statistically consistent algorithm to achieve label savings over using $O$ alone under all circumstances. For example, if labels provided by $W$ are the complete opposite of $O$, no algorithm will achieve both consistency and label savings. We next provide an assumption under which Algorithm 1 works and yields label savings.

**Assumption.** The following assumption states that difference hypothesis class contains a good cost-sensitive predictor of when $O$ and $W$ differ in the disagreement region of $\mathrm{B}_U(h^*, r)$; a predictor is good if it has low false-negative error and predicts a positive label with low frequency. If there is no such predictor, then we cannot expect an algorithm similar to ours to achieve label savings.

**Assumption 1.** *Let $\mathscr{D}$ be the joint distribution:* $\mathbb{P}_{\mathscr{D}}(x, y_O, y_W) = \mathbb{P}_U(x)\mathbb{P}_W(y_W|x)\mathbb{P}_O(y_O|x)$. *For any $r, \eta > 0$, there exists an $h^{df}_{\eta,r} \in \mathcal{H}^{df}$ with the following properties:*

$$\mathbb{P}_{\mathscr{D}}(h^{df}_{\eta,r}(x) = -1, x \in DIS(B_U(h^*, r)), y_O \neq y_W) \leq \eta \tag{3}$$

$$\mathbb{P}_{\mathscr{D}}(h^{df}_{\eta,r}(x) = 1, x \in DIS(B_U(h^*, r))) \leq \alpha(r, \eta) \tag{4}$$

Note that (3), which states there is a $h^{df} \in \mathcal{H}^{df}$ with low false-negative error, is minimally restrictive, and is trivially satisfied if $\mathcal{H}^{df}$ includes the constant classifier that always predicts 1. Theorem shows that (3) is sufficient to ensure statistical consistency.

(4) in addition states that the number of positives predicted by the classifier $h^{df}_{\eta,r}$ is upper bounded by $\alpha(r, \eta)$. Note $\alpha(r, \eta) \leq \mathbb{P}_U(\mathrm{DIS}(\mathrm{B}_U(h^*, r)))$ always; performance gain is obtained when $\alpha(r, \eta)$ is lower, which happens when the difference classifier predicts agreement on a significant portion of $\mathrm{DIS}(\mathrm{B}_U(h^*, r))$.

**Consistency.** Provided Assumption 1 holds, we next show that Algorithm 1 is statistically consistent. Establishing consistency is non-trivial for our algorithm as the output classifier is trained on labels from both $O$ and $W$.

**Theorem 1** (Consistency). *Let $h^*$ be the classifier that minimizes the error with respect to D. If Assumption 1 holds, then with probability $\geq 1 - \delta$, the classifier $\hat{h}$ output by Algorithm 1 satisfies: $err_D(\hat{h}) \leq err_D(h^*) + \varepsilon$.*

**Label Complexity.** The label complexity of standard DBAL is measured in terms of the disagreement coefficient. The disagreement coefficient $\theta(r)$ at scale $r$ is defined as: $\theta(r) = \sup_{h \in \mathscr{H}} \sup_{r' \geq r} \frac{\mathbb{P}_U(\text{DIS}(B_U(h,r')))}{r'}$; intuitively, this measures the rate of shrinkage of the disagreement region with the radius of the ball $B_U(h,r)$ for any $h$ in $\mathscr{H}$. It was shown by [9] that the label complexity of DBAL for target excess generalization error $\varepsilon$ is $\tilde{O}(d\theta(2\nu + \varepsilon)(1 + \frac{\nu^2}{\varepsilon^2}))$ where the $\tilde{O}$ notation hides factors logarithmic in $1/\varepsilon$ and $1/\delta$. In contrast, the label complexity of our algorithm can be stated in Theorem 2. Here we use the $\tilde{O}$ notation for convenience; we have the same dependence on $\log 1/\varepsilon$ and $\log 1/\delta$ as the bounds for DBAL.

**Theorem 2** (Label Complexity). *Let $d$ be the VC dimension of $\mathscr{H}$ and let $d'$ be the VC dimension of $\mathscr{H}^{df}$. If Assumption 1 holds, and if the error of the best classifier in $\mathscr{H}$ on D is $\nu$, then with probability $\geq 1 - \delta$, the following hold:*

*1. The number of label queries made by Algorithm 1 to the oracle O in epoch k at most:*

$$m_k = \tilde{O}\left( \frac{d(2\nu + \varepsilon_{k-1})(\alpha(2\nu + \varepsilon_{k-1}, \frac{\varepsilon_{k-1}}{1024}) + \varepsilon_{k-1})}{\varepsilon_k^2} + \frac{d'\mathbb{P}(DIS(B_U(h^*, 2\nu + \varepsilon_{k-1})))}{\varepsilon_k} \right) \quad (5)$$

*2. The total number of label queries made by Algorithm 1 to the oracle O is at most:*

$$\tilde{O}\left( \sup_{r \geq \varepsilon} \frac{\alpha(2\nu + r, \frac{r}{1024}) + r}{2\nu + r} \cdot d\left( \frac{\nu^2}{\varepsilon^2} + 1 \right) + \theta(2\nu + \varepsilon)d'\left( \frac{\nu}{\varepsilon} + 1 \right) \right) \quad (6)$$

### 4.1 Discussion

The first terms in (5) and (6) represent the labels needed to learn the target classifier, and second terms represent the overhead in learning the difference classifier.

In the realistic agnostic case (where $\nu > 0$), as $\varepsilon \to 0$, the second terms are *lower order* compared to the label complexity of DBAL. Thus *even if $d'$ is somewhat larger than $d$, fitting the difference classifier does not incur an asymptotically high overhead in the more realistic agnostic case.* In the realizable case, when $d' \approx d$, the second terms are of the same order as the first; therefore we should use a simpler difference hypothesis class $\mathscr{H}^{df}$ in this case. We believe that the lower order overhead term comes from the fact that there exists a classifier in $\mathscr{H}^{df}$ whose false negative error is very low.

Comparing Theorem 2 with the corresponding results for DBAL, we observe that instead of $\theta(2\nu + \varepsilon)$, we have the term $\sup_{r \geq \varepsilon} \frac{\alpha(2\nu + r, r/1024)}{2\nu + r}$. Since $\sup_{r \geq \varepsilon} \frac{\alpha(2\nu + r, r/1024)}{2\nu + r} \leq \theta(2\nu + \varepsilon)$, the *worst case* asymptotic label complexity is the same as that of standard DBAL. This label complexity may be considerably better however if $\sup_{r \geq \varepsilon} \frac{\alpha(2\nu + r, r/1024)}{2\nu + r}$ is less than the disagreement coefficient. As we expect, this will happen when the region of difference between $W$ and $O$ restricted to the disagreement regions is relatively small, and this region is well-modeled by the difference hypothesis class $\mathscr{H}^{df}$.

An interesting case is when the weak labeler differs from $O$ close to the decision boundary and agrees with $O$ away from this boundary. In this case, any consistent algorithm should switch to querying $O$ close to the decision boundary. Indeed in earlier epochs, $\alpha$ is low, and our algorithm obtains a good difference classifier and achieves label savings. In later epochs, $\alpha$ is high, the difference classifiers always predict a difference and the label complexity of the later epochs of our algorithm is the same order as DBAL. In practice, if we suspect that we are in this case, we can switch to plain active learning once $\varepsilon_k$ is small enough.

**Case Study: Linear Classfication under Uniform Distribution.** We provide a simple example where our algorithm provides a better asymptotic label complexity than DBAL. Let $\mathscr{H}$ be the class

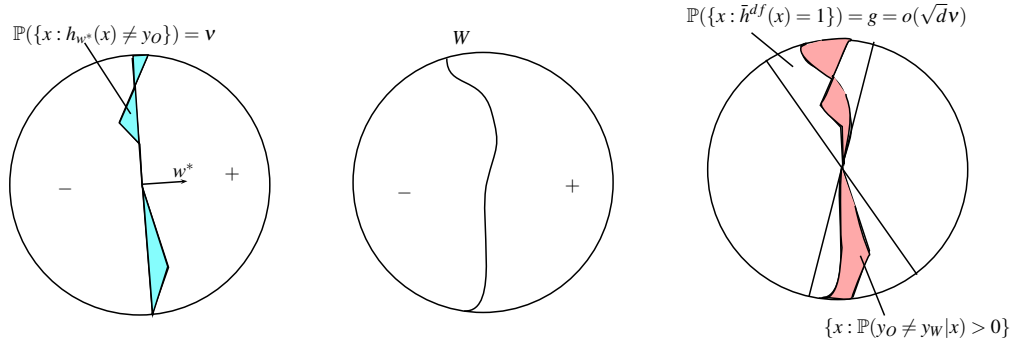

Figure 1: Linear classification over unit ball with $d = 2$. Left: Decision boundary of labeler $O$ and $h^* = h_{w^*}$. The region where $O$ differs from $h^*$ is shaded, and has probability $\nu$. Middle: Decision boundary of weak labeler $W$. Right: $\bar{h}^{df}$, $W$ and $O$. Note that $\{x : \mathbb{P}(y_O \neq y_W | x) > 0\} \subseteq \{x : \bar{h}^{df}(x) = 1\}$.

of homogeneous linear separators on the $d$-dimensional unit ball and let $\mathscr{H}^{df} = \{h\Delta h' : h, h' \in \mathscr{H}\}$. Furthermore, let $U$ be the uniform distribution over the unit ball.

Suppose that $O$ is a deterministic labeler such that $\text{err}_D(h^*) = \nu > 0$. Moreover, suppose that $W$ is such that there exists a difference classifier $\bar{h}^{df}$ with false negative error 0 for which $\mathbb{P}_U(\bar{h}^{df}(x) = 1) \leq g$. Additionally, we assume that $g = o(\sqrt{d}\nu)$; observe that this is not a strict assumption on $\mathscr{H}^{df}$, as $\nu$ could be as much as a constant. Figure 1 shows an example in $d = 2$ that satisfies these assumptions. In this case, as $\varepsilon \to 0$, Theorem 2 gives the following label complexity bound.

**Corollary 1.** *With probability $\geq 1 - \delta$, the number of label queries made to oracle $O$ by Algorithm 1 is $\tilde{O}\left(d\max(\frac{g}{\nu}, 1)(\frac{\nu^2}{\varepsilon^2} + 1) + d^{3/2}\left(1 + \frac{\nu}{\varepsilon}\right)\right)$, where the $\tilde{O}$ notation hides factors logarithmic in $1/\varepsilon$ and $1/\delta$.*

As $g = o(\sqrt{d}\nu)$, this improves over the label complexity of DBAL, which is $\tilde{O}(d^{3/2}(1 + \frac{\nu^2}{\varepsilon^2}))$.

**Conclusion.** In this paper, we take a step towards a theoretical understanding of active learning from multiple annotators through a learning theoretic formalization for learning from weak and strong labelers. Our work shows that multiple annotators can be successfully combined to do active learning in a statistically consistent manner under a general setting with few assumptions; moreover, under reasonable conditions, this kind of learning can provide label savings over plain active learning.

An avenue for future work is to explore a more general setting where we have multiple labelers with expertise on different regions of the input space. Can we combine inputs from such labelers in a statistically consistent manner? Second, our algorithm is intended for a setting where $W$ is biased, and performs suboptimally when the label generated by $W$ is a random corruption of the label provided by $O$. How can we account for both random noise and bias in active learning from weak and strong labelers?

### Acknowledgements

We thank NSF under IIS 1162581 for research support and Jennifer Dy for introducing us to the problem of active learning from multiple labelers.

## Footnotes

[1]Note that if in Algorithm 3, the upper confidence bound of $\mathbb{P}_{x \sim U}(\textbf{in\_disagr\_region}(\hat{T}, \frac{3\varepsilon}{2}, x) = 1)$ is lower than $\varepsilon/64$, then we can halt Algorithm 2 and return an arbitrary $h^{df}$ in $\mathcal{H}^{df}$. Using this $h^{df}$ will still guarantee the correctness of Algorithm 1.

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
