[Supplementary Material]

# A   Notation

## A.1   Basic Definitions and Notation

Here we do a brief recap of notation. We assume that we are given a target hypothesis class $\mathcal{H}$ of VC dimension $d$, and a difference hypothesis class $\mathcal{H}^{df}$ of VC dimension $d'$.

We are given access to an unlabeled distribution $U$ and two labeling oracles $O$ and $W$. Querying $O$ (resp. $W$) with an unlabeled data point $x_i$ generates a label $y_{i,O}$ (resp. $y_{i,W}$) which is drawn from the distribution $\mathbb{P}_O(y|x_i)$ (resp. $\mathbb{P}_W(y|x_i)$). In general these two distributions are different. We use the notation $\mathcal{D}$ to denote the joint distribution over examples and labels from $O$ and $W$:

$$\mathbb{P}_{\mathcal{D}}(x, y_O, y_W) = \mathbb{P}_U(x)\mathbb{P}_O(y_O|x)\mathbb{P}_W(y_W|x)$$

Our goal in this paper is to learn a classifier in $\mathcal{H}$ which has low error with respect to the data distribution $D$ described as: $\mathbb{P}_D(x,y) = \mathbb{P}_U(x)\mathbb{P}_O(y|x)$ and our goal is use queries to $W$ to reduce the number of queries to $O$. We use $y_O$ to denote the labels returned by $O$, $y_W$ to denote the labels returned by $W$.

The error of a classifier $h$ under a labeled data distribution $Q$ is defined as: $\mathrm{err}_Q(h) = \mathbb{P}_{(x,y)\sim Q}(h(x) \neq y)$; we use the notation $\mathrm{err}(h, S)$ to denote its empirical error on a labeled data set $S$. We use the notation $h^*$ to denote the classifier with the lowest error under $D$. Define the excess error of $h$ with respect to distribution $D$ as $\mathrm{err}_D(h) - \mathrm{err}_D(h^*)$. For a set $Z$, we occasionally abuse notation and use $Z$ to also denote the uniform distribution over the elements of $Z$.

**Confidence Sets and Disagreement Region.**   Our active learning algorithm will maintain a $(1-\delta)$-*confidence set* for $h^*$ throughout the algorithm. A set of classifiers $V \subseteq \mathcal{H}$ produced by a (possibly randomized) algorithm is said to be a $(1-\delta)$-confidence set for $h^*$ if $h^* \in V$ with probability $\geq 1-\delta$; here the probability is over the randomness of the algorithm as well as the choice of all labeled and unlabeled examples drawn by it.

Given two classifiers $h_1$ and $h_2$ the disagreement between $h_1$ and $h_2$ under an unlabeled data distribution $U$, denoted by $\rho_U(h_1, h_2)$, is $\mathbb{P}_{x\sim U}(h_1(x) \neq h_2(x))$. Given an unlabeled dataset $S$, the empirical disagreement of $h_1$ and $h_2$ on $S$ is denoted by $\rho_S(h_1, h_2)$. Observe that the disagreements under $U$ form a pseudometric over $\mathcal{H}$. We use $\mathrm{B}_U(h, r)$ to denote a ball of radius $r$ centered around $h$ in this metric. The *disagreement region* of a set $V$ of classifiers, denoted by $\mathrm{DIS}(V)$, is the set of all examples $x \in \mathcal{X}$ such that there exist two classifiers $h_1$ and $h_2$ in $V$ for which $h_1(x) \neq h_2(x)$.

**Disagreement Region.**   We denote the disagreement region of a disagreement ball of radius $r$ centered around $h^*$ by

$$\Delta(r) := \mathrm{DIS}(\mathrm{B}(h^*, r)) \tag{7}$$

**Concentration Inequalities.**   Suppose $Z$ is a dataset consisting of $n$ iid samples from a distribution $D$. We will use the following result, which is obtained from a standard application of the normalized VC inequality. With probability $1-\delta$ over the random draw of $Z$, for all $h, h' \in \mathcal{H}$,

$$\begin{aligned}
&|(\mathrm{err}(h,Z) - \mathrm{err}(h',Z)) - (\mathrm{err}_D(h) - \mathrm{err}_D(h'))| \\
&\leq \quad \min\left(\sqrt{\sigma(n,\delta)\rho_Z(h,h')} + \sigma(n,\delta), \sqrt{\sigma(n,\delta)\rho_D(h,h')} + \sigma(n,\delta)\right)
\end{aligned} \tag{8}$$

$$\begin{aligned}
&|(\mathrm{err}(h,Z) - \mathrm{err}_D(h)| \\
&\leq \quad \min\left(\sqrt{\sigma(n,\delta)\mathrm{err}(h,Z)} + \sigma(n,\delta), \sqrt{\sigma(n,\delta)\mathrm{err}_D(h)} + \sigma(n,\delta)\right)
\end{aligned} \tag{9}$$

where $d$ is the VC dimension of $\mathcal{H}$ and the notation $\sigma(n,\delta)$ is defined as:

$$\sigma(n,\delta) = \frac{8}{n}\left(2d\ln\frac{2en}{d} + \ln\frac{24}{\delta}\right) \tag{10}$$

Equation (8) loosely implies the following equation:

$$|(\mathrm{err}(h,Z) - \mathrm{err}(h',Z)) - (\mathrm{err}_D(h) - \mathrm{err}_D(h'))| \leq \sqrt{4\sigma(n,\delta)} \tag{11}$$

The following is a consequence of standard Chernoff bounds. Let $X_1, \ldots, X_n$ be iid Bernoulli random variables with mean $p$. If $\hat{p} = \sum_i X_i / n$, then with probabiliy $1 - \delta$,

$$|\hat{p} - p| \leq \min(\sqrt{p\gamma(n,\delta)} + \gamma(n,\delta), \sqrt{\hat{p}\gamma(n,\delta)} + \gamma(n,\delta)) \quad (12)$$

where the notation $\gamma(n,\delta)$ is defined as:

$$\gamma(n,\delta) = \frac{4}{n} \ln \frac{2}{\delta} \quad (13)$$

Equation (12) loosely implies the following equation:

$$|\hat{p} - p| \leq \sqrt{4\gamma(n,\delta)} \quad (14)$$

Using the notation we just introduced, we can rephrase Assumption 1 as follows. For any $r, \eta > 0$, there exists an $h_{\eta,r}^{df} \in \mathcal{H}^{df}$ with the following properties:

$$\mathbb{P}_{\mathcal{D}}(h_{\eta,r}^{df}(x) = -1, x \in \Delta(r), y_O \neq y_W) \leq \eta$$
$$\mathbb{P}_{\mathcal{D}}(h_{\eta,r}^{df}(x) = 1, x \in \Delta(r)) \leq \alpha(r,\eta)$$

We end with an useful fact about $\sigma(n,\delta)$.

**Fact 1.** *The minimum n such that $\sigma(n, \delta/(\log n(\log n + 1))) \leq \varepsilon$ is at most*

$$\frac{64}{\varepsilon}(d \ln \frac{512}{\varepsilon} + \ln \frac{24}{\delta})$$

## A.2 Adaptive Procedure for Estimating Probability Mass

For completeness, we describe in Algorithm 3 a standard doubling procedure for estimating the bias of a coin within a constant factor. This procedure is used by Algorithm 2 to estimate the probability mass of the disagreement region of the current confidence set based on unlabeled examples drawn from $U$.

---
**Algorithm 3** Adaptive Procedure for Estimating the Bias of a Coin
---
1: Input: failure probability $\delta$, an oracle $\mathcal{O}$ which returns iid Bernoulli random variables with unknown bias $p$.
2: Output: $\hat{p}$, an estimate of bias $p$ such that $\hat{p} \leq p \leq 2\hat{p}$ with probability $\geq 1 - \delta$.
3: **for** $i = 1, 2, \ldots$ **do**
4:     Call the oracle $\mathcal{O}$ $2^i$ times to get empirical frequency $\hat{p}_i$.
5:     **if** $\sqrt{\frac{4 \ln \frac{4 \cdot 2^i}{\delta}}{2^i}} \leq \hat{p}_i / 3$ **then return** $\hat{p} = \frac{2\hat{p}_i}{3}$
6:     **end if**
7: **end for**
---

**Lemma 1.** *Suppose $p > 0$ and Algorithm 3 is run with failure probability $\delta$. Then with probability $1 - \delta$, (1) the output $\hat{p}$ is such that $\hat{p} \leq p \leq 2\hat{p}$. (2) The total number of calls to $\mathcal{O}$ is at most $O(\frac{1}{p^2} \ln \frac{1}{\delta p})$.*

*Proof.* Consider the event

$$E = \{ \text{ for all } i \in \mathbb{N}, |\hat{p}_i - p| \leq \sqrt{\frac{4 \ln \frac{2 \cdot 2^i}{\delta}}{2^i}} \}$$

By Equation (14) and union bound, $\mathbb{P}(E) \geq 1 - \delta$. On event $E$, we claim that if $i$ is large enough that

$$4\sqrt{\frac{4 \ln \frac{4 \cdot 2^i}{\delta}}{2^i}} \leq p \quad (15)$$

then the condition in line 5 will be met. Indeed, this implies

$$\sqrt{\frac{4\ln\frac{4\cdot2^i}{\delta}}{2^i}} \leq \frac{p-\sqrt{\frac{4\ln\frac{4\cdot2^i}{\delta}}{2^i}}}{3} \leq \frac{\hat{p}_i}{3}$$

Define $i_0$ as the smallest number $i$ such that Equation (15) is true. Then by algebra, $2^{i_0} = O(\frac{1}{p^2}\ln\frac{1}{\delta p})$. Hence the number of calls to oracle $\mathscr{O}$ is at most $1+2+\ldots+2^{i_0} = O(\frac{1}{p^2}\ln\frac{1}{\delta p})$.
Consider the smallest $i^*$ such that the condition in line 5 is met. We have that

$$\sqrt{\frac{4\ln\frac{4\cdot2^{i^*}}{\delta}}{2^{i^*}}} \leq \hat{p}_{i^*}/3$$

By the definition of $E$,
$$|p-\hat{p}_{i^*}| \leq \hat{p}_{i^*}/3$$
that is, $2\hat{p}_{i^*}/3 \leq p \leq 4\hat{p}_{i^*}/3$, implying $\hat{p} \leq p \leq 2\hat{p}$. □

## A.3 Notations on Datasets

Without loss of generality, assume the examples drawn throughout Algorithm 1 have distinct feature values $x$, since this happens with probability 1 under mild assumptions.

Algorithm 1 uses a mixture of three kinds of labeled data to learn a target classifier – labels obtained from querying $O$, labels inferred by the algorithm, and labels obtained from querying $W$. To analyze the effect of these three kinds of labeled data, we need to introduce some notation.

Recall that we define the joint distribution $\mathscr{D}$ over examples and labels both from $O$ and $W$ as follows:
$$\mathbb{P}_{\mathscr{D}}(x,y_O,y_W) = \mathbb{P}_U(x)\mathbb{P}_O(y_O|x)\mathbb{P}_W(y_W|x)$$
where given an example $x$, the labels generated by $O$ and $W$ are conditionally independent.

A dataset $\hat{S}$ with empirical error minimizer $\hat{h}$ and a rejection threshold $\tau$ define a implicit confidence set for $h^*$ as follows:
$$V(\hat{S},\tau) = \{h : \text{err}(h,\hat{S}) - \text{err}(\hat{h},\hat{S}) \leq \tau\}$$
At the beginning of epoch $k$, we have $\hat{S}_{k-1}$. $\hat{h}_{k-1}$ is defined as the empirical error minimizer of $\hat{S}_{k-1}$. The disagreement region of the implicit confidence set at epoch $k$, $R_{k-1}$ is defined as $R_{k-1} := \text{DIS}(V(\hat{S}_{k-1}, 3\varepsilon_k/2))$. Algorithm 4 **in_disagr_region**$(\hat{S}_{k-1}, 3\varepsilon_k/2, x)$ provides a test deciding if an unlabeled example $x$ is inside $R_{k-1}$ in epoch $k$. (See Lemma 6.)

Define $\mathscr{A}_k$ to be the distribution $\mathscr{D}$ conditioned on the set $\{(x,y_O,y_W) : x \in R_{k-1}\}$. At epoch $k$, Algorithm 2 has inputs distribution $U$, oracles $W$ and $O$, target false negative error $\varepsilon = \varepsilon_k/128$, hypothesis class $\mathscr{H}^{df}$, confidence $\delta = \delta_k/2$, previous labeled dataset $\hat{S}_{k-1}$, and outputs a difference classifier $\hat{h}_k^{df}$. By the setting of $m$ in Equation (1), Algorithm 2 first computes $\hat{p}_k$ using unlabeled examples drawn from $U$, which is an estimator of $\mathbb{P}_{\mathscr{D}}(x \in R_{k-1})$. Then it draws a subsample of size

$$m_{k,1} = \frac{64\cdot1024\hat{p}_k}{\varepsilon_k}\left(d\ln\frac{512\cdot1024\hat{p}_k}{\varepsilon_k} + \ln\frac{144}{\delta_k}\right) \tag{16}$$

iid from $\mathscr{A}_k$. We call the resulting dataset $\mathscr{A}_k'$.

At epoch $k$, Algorithm 5 performs adaptive subsampling to refine the implicit $(1-\delta)$-confidence set. For each round $t$, it subsamples $U$ to get an unlabeled dataset $S_k^{t,U}$ of size $2^t$. Define the corresponding (hypothetical) dataset with labels queried from both $W$ and $O$ as $\mathscr{S}_k^t$. $S_k^t$, the (hypothetical) dataset with labels queried from $O$, is defined as:

$$S_k^t = \{(x,y_O)|(x,y_O,y_W) \in \mathscr{S}_k^t\}$$

In addition to obtaining labels from $O$, the algorithm obtains labels in two other ways. First, if an $x \in \mathscr{X} \setminus R_{k-1}$, then its label is safely inferred and with high probability, this inferred label $\hat{h}_{k-1}(x)$ is equal to $h^*(x)$. Second, if an $x$ lies in $R_{k-1}$ but if the difference classifier $\hat{h}_k^{df}$ predicts agreement

between $O$ and $W$, then its label is obtained by querying $W$. The actual dataset $\hat{S}_k^t$ generated by Algorithm 5 is defined as:

$$
\begin{aligned}
\hat{S}_k^t &= \{(x,\hat{h}_{k-1}(x))|(x,y_O,y_W) \in \mathscr{S}_k^t, x \notin R_{k-1}\} \cup \{(x,y_O)|(x,y_O,y_W) \in \mathscr{S}_k^t, x \in R_{k-1}, \hat{h}_k^{df}(x) = +1\} \\
&\quad \cup \{(x,y_W)|(x,y_O,y_W) \in \mathscr{S}_k^t, x \in R_{k-1}, \hat{h}_k^{df}(x) = -1\}
\end{aligned}
$$

We use $\hat{D}_k$ to denote the labeled data distribution as follows:

$$
\mathbb{P}_{\hat{D}_k}(x,y) = \mathbb{P}_U(x)\mathbb{P}_{\hat{Q}_k}(y|x)
$$

$$
\mathbb{P}_{\hat{Q}_k}(y|x) = \begin{cases} I(\hat{h}_{k-1}(x) = y), & x \notin R_{k-1} \\ \mathbb{P}_O(y|x), & x \in R_{k-1}, \hat{h}_k^{df}(x) = +1 \\ \mathbb{P}_W(y|x), & x \in R_{k-1}, \hat{h}_k^{df}(x) = -1 \end{cases}
$$

Therefore, $\hat{S}_k^t$ can be seen as a sample of size $2^t$ drawn iid from $\hat{D}_k$.

Observe that $\hat{h}_k^t$ is obtained by training an ERM classifier over $\hat{S}_k^t$, and $\delta_k^t = \delta_k/2t(t+1)$.

Suppose Algorithm 5 stops at iteration $t_0(k)$, then the final dataset returned is $\hat{S}_k = \hat{S}_k^{t_0(k)}$, with a total number of $m_{k,2}$ label requests to $O$. We define $S_k = S_k^{t_0(k)}$, $\mathscr{S}_k = \mathscr{S}_k^{t_0(k)}$ and $\sigma_k = \sigma(2^{t_0(k)}, \delta_k^{t_0(k)})$.

For $k = 0$, we define the notation $\hat{S}_k$ differently. $\hat{S}_0$ is the dataset drawn iid at random from $D$, with labels queried entirely to $O$. For notational convenience, define $S_0 = \hat{S}_0$. $\sigma_0$ is defined as $\sigma_0 = \sigma(n_0, \delta_0)$, where $\sigma(\cdot, \cdot)$ is defined by Equation (10) and $n_0$ is defined as:

$$
n_0 = (64 \cdot 1024^2)(2d\ln(512 \cdot 1024^2) + \ln\frac{96}{\delta})
$$

Recall that $\hat{h}_k = \operatorname{argmin}_{h \in \mathscr{H}} \operatorname{err}(h, \hat{S}_k)$ is the empirical error minimizer with respect to the dataset $\hat{S}_k$.

Note that the empirical distance $\rho_Z(\cdot, \cdot)$ does not depend on the labels in dataset $Z$, therefore, $\rho_{\hat{S}_k}(h,h') = \rho_{S_k}(h,h')$. We will use them interchangably throughout.

## A.4 Events

Recall that $\delta_k = \delta/(4(k+1)^2), \varepsilon_k = 2^{-k}$.

Define

$$
h_k^{df} = h_{2v+\varepsilon_{k-1}, \varepsilon_k/512}^{df}
$$

where the notation $h_{r,\eta}^{df}$ is introduced in Assumption 1.

We begin by defining some events that we will condition on later in the proof, and showing that these events occur with high probability.

Define event

$$
\begin{aligned}
E_k^1 := \Big\{ \quad & \mathbb{P}_{\mathscr{D}}(x \in R_{k-1})/2 \le \hat{p}_k \le \mathbb{P}_{\mathscr{D}}(x \in R_{k-1}), \\
& \text{and For all } h^{df} \in \mathscr{H}^{df}, \\
& |\mathbb{P}_{\mathscr{A}_k'}(h^{df}(x) = -1, y_O \ne y_W) - \mathbb{P}_{\mathscr{A}_k}(h^{df}(x) = -1, y_O \ne y_W)| \le \frac{\varepsilon_k}{1024\mathbb{P}_{\mathscr{D}}(x \in R_{k-1})} \\
& + \sqrt{\min(\mathbb{P}_{\mathscr{A}_k}(h^{df}(x) = -1, y_O \ne y_W), \mathbb{P}_{\mathscr{A}_k'}(h^{df}(x) = -1, y_O \ne y_W)) \frac{\varepsilon_k}{1024\mathbb{P}_{\mathscr{D}}(x \in R_{k-1})}} \\
& \text{and } |\mathbb{P}_{\mathscr{A}_k'}(h^{df}(x) = +1) - \mathbb{P}_{\mathscr{A}_k}(h^{df}(x) = +1)| \\
\le \quad & \sqrt{\min(\mathbb{P}_{\mathscr{A}_k}(h^{df}(x) = +1), \mathbb{P}_{\mathscr{A}_k'}(h^{df}(x) = +1)) \frac{\varepsilon_k}{1024\mathbb{P}_{\mathscr{D}}(x \in R_{k-1})}} + \frac{\varepsilon_k}{1024\mathbb{P}_{\mathscr{D}}(x \in R_{k-1})} \Big\}
\end{aligned}
$$

**Fact 2.** $\mathbb{P}(E_k^1) \ge 1 - \delta_k/2$.

Table 1: Summary of Notations.

| Notation | Explanation | Samples Drawn from |
|---|---|---|
| $\mathscr{D}$ | Joint distribution of $(x, y_W, y_O)$ | - |
| $D$ | Joint distribution of $(x, y_O)$ | - |
| $U$ | Marginal distribution of $x$ | - |
| $O$ | Conditional distribution of $y_O$ given $x$ | - |
| $W$ | Conditional distribution of $y_W$ given $x$ | - |
| $R_{k-1}$ | Disagreement region at epoch $k$ | - |
| $\mathscr{A}_k$ | Conditional distribution of $(x, y_W, y_O)$ given $x \in R_{k-1}$ | - |
| $\mathscr{A}'_k$ | Dataset used to train difference classifier at epoch $k$ | $\mathscr{A}_k$ |
| $h_k^{df}$ | Difference classifier $h_{2v+\varepsilon_{k-1}, \varepsilon_k/512}^{df}$, where $h_{\eta,r}$ is defined in Assumption 1 | - |
| $\hat{h}_k^{df}$ | Difference classifier returned by Algorithm 2 at epoch $k$ | - |
| $S_k^{t,U}$ | unlabeled dataset drawn at iteration $t$ of Algorithm 5 at epoch $k \geq 1$ | $U$ |
| $\mathscr{S}_k^t$ | $S_k^{t,U}$ augmented by labels from $O$ and $W$ | $\mathscr{D}$ |
| $S_k^t$ | $\{(x, y_O) \mid (x, y_O, y_W) \in \mathscr{S}_k^t\}$ | $D$ |
| $\hat{S}_k^t$ | Labeled dataset produced at iteration $t$ of Algorithm 5 at epoch $k \geq 1$ | $\hat{D}_k$ |
| $\hat{D}_k$ | Distribution of $\hat{S}_k^t$ for $k \geq 1$ and any $t$. Has marginal $U$ over $\mathscr{X}$. The conditional distribution of $y\mid x$ is $I(h^*(x))$ if $x \notin R_{k-1}$, $W$ if $x \in R_{k-1}$ and $\hat{h}^{df}(x) = -1$, and $O$ otherwise | - |
| $t_0(k)$ | Number of iterations of Algorithm 5 at epoch $k \geq 1$ | - |
| $\hat{S}_0$ | Initial dataset drawn by Algorithm 1 | $D$ |
| $\hat{S}_k$ | Dataset finally returned by Algorithm 5 at epoch $k \geq 1$. Equal to $\hat{S}_k^{t_0(k)}$ | $\hat{D}_k$ |
| $S_k$ | Dataset obtained by replacing all labels in $\hat{S}_k$ by labels drawn from $O$. Equal to $S_k^{t_0(k)}$ | $D$ |
| $\mathscr{S}_k$ | Equal to $\mathscr{S}_k^{t_0(k)}$ | $\mathscr{D}$ |
| $\hat{h}_k$ | Empirical error minimizer on $\hat{S}_k$ | - |

Define event

$$
E_k^2 = \Big\{ \quad \text{For all } t \in \mathbb{N}, \text{ for all } h, h' \in \mathscr{H},
$$

$$
|(\text{err}(h, S_k^t) - \text{err}(h', S_k^t)) - (\text{err}_D(h) - \text{err}_D(h'))| \leq \sigma(2^t, \delta_k^t) + \sqrt{\sigma(2^t, \delta_k^t)\rho_{S_k^t}(h, h')}
$$

and $\quad \text{err}(h, \hat{S}_k^t) - \text{err}_{\hat{D}_k}(h) \leq \sigma(2^t, \delta_k^t) + \sqrt{\sigma(2^t, \delta_k^t)\text{err}_{\hat{D}_k}(h)}$

and $\quad \mathbb{P}_{\mathscr{S}_k^t}(\hat{h}_k^{df}(x) = -1, y_O \neq y_W, x \in R_{k-1}) - \mathbb{P}_{\mathscr{D}}(\hat{h}_k^{df}(x) = -1, y_O \neq y_W, x \in R_{k-1})$

$$
\leq \sqrt{\gamma(2^t, \delta_k^t)\mathbb{P}_{\mathscr{S}_k^t}(\hat{h}_k^{df}(x) = -1, y_O \neq y_W, x \in R_{k-1})} + \gamma(2^t, \delta_k^t)
$$

and $\quad \mathbb{P}_{\mathscr{S}_k^t}(\hat{h}_k^{df}(x) = -1 \cap x \in R_{k-1}) \leq 2(\mathbb{P}_{\mathscr{D}}(\hat{h}_k^{df}(x) = -1, x \in R_{k-1}) + \gamma(2^t, \delta_k^t))\Big\}$

**Fact 3.** $\mathbb{P}(E_k^2) \geq 1 - \delta_k/2$.

We will also use the following definitions of events in our proof. Define event $F_0$ as

$$
F_0 = \Big\{ \text{for all } h, h' \in \mathscr{H}, |(\text{err}(h, S_0) - \text{err}(h', S_0)) - (\text{err}_D(h) - \text{err}_D(h'))| \leq \sigma(n_0, \delta_0) + \sqrt{\sigma(n_0, \delta_0)\rho_{S_0}(h, h')} \Big\}
$$

For $k \in \{1, 2, \ldots, k_0\}$, event $F_k$ is defined inductively as

$$F_k = F_{k-1} \cap (E_k^1 \cap E_k^2)$$

**Fact 4.** *For $k \in \{0, 1, \ldots, k_0\}$, $\mathbb{P}(F_k) \geq 1 - \delta_0 - \delta_1 - \ldots - \delta_k$. Specifically, $\mathbb{P}(F_{k_0}) \geq 1 - \delta$.*

The proofs of Facts 2, 3 and 4 are provided in Appendix E.

# B   Proof Outline and Main Lemmas

The main idea of the proof is to maintain the following three invariants on the outputs of Algorithm 1 in each epoch. We prove that these invariants hold simultaneously for each epoch with high probability by induction over the epochs. Throughout, for $k \geq 1$, the end of epoch $k$ refers to the end of execution of line 13 of Algorithm 1 at iteration $k$. The end of epoch 0 refers to the end of execution of line 5 in Algorithm 1.

Invariant 1 states that if we replace the inferred labels and labels obtained from $W$ in $\hat{S}_k$ by those obtained from $O$ (thus getting the dataset $S_k$), then the excess errors of classifiers in $\mathcal{H}$ will not decrease by much.

**Invariant 1** (Approximate Favorable Bias). *Let $h$ be any classifier in $\mathcal{H}$, and $h'$ be another classifier in $\mathcal{H}$ with excess error on $D$ no greater than $\varepsilon_k$. Then, at the end of epoch $k$, we have:*

$$err(h, S_k) - err(h', S_k) \leq err(h, \hat{S}_k) - err(h', \hat{S}_k) + \varepsilon_k/16$$

Invariant 2 establishes that in epoch $k$, Algorithm 5 selects enough examples so as to ensure that concentration of empirical errors of classifiers in $\mathcal{H}$ on $S_k$ to their true errors.

**Invariant 2** (Concentration). *At the end of epoch $k$, $\hat{S}_k$, $S_k$ and $\sigma_k$ are such that:*
*1. For any pair of classifiers $h, h' \in \mathcal{H}$, it holds that:*

$$|(err(h, S_k) - err(h', S_k)) - (err_D(h) - err_D(h'))| \leq \sigma_k + \sqrt{\sigma_k \rho_{S_k}(h, h')} \qquad (17)$$

*2. The dataset $\hat{S}_k$ has the following property:*

$$\sigma_k + \sqrt{\sigma_k err(\hat{h}_k, \hat{S}_k)} \leq \varepsilon_k/512 \qquad (18)$$

Finally, Invariant 3 ensures that the difference classifier produced in epoch $k$ has low false negative error on the disagreement region of the $(1 - \delta)$ confidence set at epoch $k$.

**Invariant 3** (Difference Classifier). *At epoch $k$, the difference classifier output by Algorithm 2 is such that*

$$\mathbb{P}_{\mathscr{D}}(\hat{h}_k^{df}(x) = -1, y_O \neq y_W, x \in R_{k-1}) \leq \varepsilon_k/64 \qquad (19)$$

$$\mathbb{P}_{\mathscr{D}}(\hat{h}_k^{df}(x) = +1, x \in R_{k-1}) \leq 6(\alpha(2\nu + \varepsilon_{k-1}, \varepsilon_k/512) + \varepsilon_k/1024) \qquad (20)$$

We will show the following property about the three invariants. Its proof is deferred to Subsection B.4.

**Lemma 2.** *There is a numerical constant $c_0 > 0$ such that the following holds. The collection of events $\{F_k\}_{k=0}^{k_0}$ is such that for $k \in \{0, 1, \ldots, k_0\}$:*
*(1) If $k = 0$, then on event $F_k$, at epoch $k$,*
*(1.1) Invariants 1,2 hold.*
*(1.2) The number of label requests to $O$ is at most $m_0 \leq c_0(d + \ln \frac{1}{\delta})$.*
*(2) If $k \geq 1$, then on event $F_k$, at epoch $k$,*
*(2.1) Invariants 1,2,3 hold.*
*(2.2) the number of label requests to $O$ is at most*

$$m_k \leq c_0 \left( \frac{(\alpha(2\nu + \varepsilon_{k-1}, \varepsilon_k/1024) + \varepsilon_k)(\nu + \varepsilon_k)}{\varepsilon_k^2} d(\ln^2 \frac{1}{\varepsilon_k} + \ln^2 \frac{1}{\delta_k}) + \frac{\mathbb{P}_U(x \in \Delta(2\nu + \varepsilon_{k-1}))}{\varepsilon_k}(d' \ln \frac{1}{\varepsilon_k} + \ln \frac{1}{\delta_k}) \right)$$

---

**Algorithm 4 in_disagr_region**$(\hat{S},\tau,x)$: Test if $x$ is in the disagreement region of current confidence set

---

 1: Input: labeled dataset $\hat{S}$, rejection threshold $\tau$, unlabeled example $x$.
 2: Output: 1 if $x$ in the disagreement region of current confidence set, 0 otherwise.
 3: Train $\hat{h} \leftarrow$ CONS-LEARN$_{\mathscr{H}}(\{\emptyset,\hat{S}\})$.
 4: Train $\hat{h}'_x \leftarrow$ CONS-LEARN$_{\mathscr{H}}(\{(x,-\hat{h}(x))\},\hat{S}\})$.
 5: **if** err$(\hat{h}'_x,\hat{S})-$err$(\hat{h},\hat{S}) > \tau$ **then**     *# x is in the agreement region*
 6:     **return** 0
 7: **else**   *# x is in the disagreement region*
 8:     **return** 1
 9: **end if**

---

## B.1   Active Label Inference and Identifying the Disagreement Region

We begin by proving some lemmas about Algorithm 4 which identifies if an example lies in the disagreement region of the current confidence set. This is done by using a constrained ERM oracle CONS-LEARN$_H(\cdot,\cdot)$ using ideas similar to [9, 14, 3, 4].

**Lemma 3.** *When given as input a dataset $\hat{S}$, a threshold $\tau > 0$, an unlabeled example x, Algorithm 4* **in_disagr_region** *returns 1 if and only if x lies inside $DIS(V(\hat{S},\tau))$.*

*Proof.* ($\Rightarrow$) If Algorithm 4 returns 1, then we have found a classifier $\hat{h}'_x$ such that (1) $\hat{h}_x(x) = -\hat{h}(x)$, and (2) err$(\hat{h}'_x,\hat{S})-$err$(\hat{h},\hat{S}) \le \tau$, i.e. $\hat{h}'_x \in V(\hat{S},\tau)$. Therefore, $x$ is in DIS$(V(\hat{S},\tau))$.
($\Leftarrow$) If $x$ is in DIS$(V(\hat{S},\tau))$, then there exists a classifier $h \in \mathscr{H}$ such that (1) $h(x) = -\hat{h}(x)$ and (2) err$(h,\hat{S})-$err$(\hat{h},\hat{S}) \le \tau$. Hence by definition of $\hat{h}'_x$, err$(\hat{h}'_x,\hat{S})-$err$(\hat{h},\hat{S}) \le \tau$. Thus, Algorithm 4 returns 1. $\qquad\square$

We now provide some lemmas about the behavior of Algorithm 4 called at epoch $k$.

**Lemma 4.** *Suppose Invariants 1 and 2 hold at the end of epoch $k-1$. If $h \in \mathscr{H}$ is such that $err_D(h) \le err_D(h^*) + \varepsilon_{k-1}/2$, then*

$$err(h,\hat{S}_{k-1}) - err(\hat{h}_{k-1},\hat{S}_{k-1}) \le 3\varepsilon_{k-1}/4$$

*Proof.* If $h \in \mathscr{H}$ has excess error at most $\varepsilon_{k-1}/2$ with respect to $D$, then,

$$
\begin{aligned}
&\text{err}(h,\hat{S}_{k-1}) - \text{err}(\hat{h}_{k-1},\hat{S}_{k-1}) \\
\le\ & \text{err}(h,S_{k-1}) - \text{err}(\hat{h}_{k-1},S_{k-1}) + \varepsilon_{k-1}/16 \\
\le\ & \text{err}_D(h) - \text{err}_D(\hat{h}_{k-1}) + \sigma_{k-1} + \sqrt{\sigma_{k-1}\rho_{S_{k-1}}(h,\hat{h}_{k-1})} + \varepsilon_{k-1}/16 \\
\le\ & \varepsilon_{k-1}/2 + \sigma_{k-1} + \sqrt{\sigma_{k-1}\rho_{S_{k-1}}(h,\hat{h}_{k-1})} + \varepsilon_{k-1}/16 \\
\le\ & 9\varepsilon_{k-1}/16 + \sigma_{k-1} + \sqrt{\sigma_{k-1}\text{err}(h,\hat{S}_{k-1})} + \sqrt{\sigma_{k-1}\text{err}(\hat{h}_{k-1},\hat{S}_{k-1})} \\
\le\ & 9\varepsilon_{k-1}/16 + \sigma_{k-1} + \sqrt{\sigma_{k-1}\text{err}(h,\hat{S}_{k-1})} + \sqrt{\sigma_{k-1}(\text{err}(\hat{h}_{k-1},\hat{S}_{k-1})+9\varepsilon_{k-1}/16)}
\end{aligned}
$$

Where the first inequality follows from Invariant 1, the second inequality from Equation (17) of Invariant 2, the third inequality from the assumption that $h$ has excess error at most $\varepsilon_{k-1}/2$, and the fourth inequality from the triangle inequality, the fifth inequality is by adding a nonnegative number

in the last term. Continuing,

$$\mathrm{err}(h, \hat{S}_{k-1}) - \mathrm{err}(\hat{h}_{k-1}, \hat{S}_{k-1})$$

$$\leq \quad 9\varepsilon_{k-1}/16 + 4\sigma_{k-1} + 2\sqrt{\sigma_{k-1}(\mathrm{err}(\hat{h}_{k-1}, \hat{S}_{k-1}) + 9\varepsilon_{k-1}/16)}$$

$$\leq \quad 9\varepsilon_{k-1}/16 + 4\sigma_{k-1} + 2\sqrt{\sigma_{k-1}\mathrm{err}(\hat{h}_{k-1}, \hat{S}_{k-1})} + 2\sqrt{\varepsilon_{k-1}/512 \cdot 9\varepsilon_{k-1}/16}$$

$$\leq \quad 9\varepsilon_{k-1}/16 + \varepsilon_{k-1}/32 + 2\sqrt{\varepsilon_{k-1}/512 \cdot 9\varepsilon_{k-1}/16}$$

$$\leq \quad 3\varepsilon_{k-1}/4$$

Where the first inequality is by simple algebra (by letting $D = \mathrm{err}(h, \hat{S}_{k-1})$, $E = \mathrm{err}(\hat{h}_{k-1}, \hat{S}_{k-1}) + 9\varepsilon_{k-1}/16$, $F = \sigma_{k-1}$ in $D \leq E + F + \sqrt{DF} + \sqrt{EF} \Rightarrow D \leq E + 4F + 2\sqrt{EF}$), the second inequality is from $\sqrt{A+B} \leq \sqrt{A} + \sqrt{B}$ and $\sigma_{k-1} \leq \varepsilon_{k-1}/512$ which utilizes Equation (18) of Invariant 2, the third inequality is again by Equation (18) of Invariant 2, the fourth inequality is by algebra. □

**Lemma 5.** *Suppose Invariants 1 and 2 hold at the end of epoch $k-1$. Then,*

$$err_D(\hat{h}_{k-1}) - err_D(h^*) \leq \varepsilon_{k-1}/8$$

*Proof.* By Lemma 4, we know that since $h^*$ has excess error 0 with respect to $D$,

$$\mathrm{err}(h^*, \hat{S}_{k-1}) - \mathrm{err}(\hat{h}_{k-1}, \hat{S}_{k-1}) \leq 3\varepsilon_{k-1}/4 \tag{21}$$

Therefore,

$$err_D(\hat{h}_{k-1}) - err_D(h^*)$$

$$\leq \quad \mathrm{err}(\hat{h}_{k-1}, S_{k-1}) - \mathrm{err}(h^*, S_{k-1}) + \sigma_{k-1} + \sqrt{\sigma_{k-1}\rho_{S_{k-1}}(\hat{h}_{k-1}, h^*)}$$

$$\leq \quad \mathrm{err}(\hat{h}_{k-1}, \hat{S}_{k-1}) - \mathrm{err}(h^*, \hat{S}_{k-1}) + \sigma_{k-1} + \sqrt{\sigma_{k-1}\rho_{S_{k-1}}(\hat{h}_{k-1}, h^*)} + \varepsilon_{k-1}/16$$

$$\leq \quad \varepsilon_{k-1}/16 + \sigma_{k-1} + \sqrt{\sigma_{k-1}(\mathrm{err}(\hat{h}_{k-1}, \hat{S}_{k-1}) + \mathrm{err}(h^*, \hat{S}_{k-1}))}$$

$$\leq \quad \varepsilon_{k-1}/16 + \sigma_{k-1} + \sqrt{\sigma_{k-1}(2\mathrm{err}(\hat{h}_{k-1}, \hat{S}_{k-1}) + 3\varepsilon_{k-1}/4)}$$

$$\leq \quad \varepsilon_{k-1}/16 + \sigma_{k-1} + \sqrt{2\sigma_{k-1}\mathrm{err}(\hat{h}_{k-1}, \hat{S}_{k-1})} + \sqrt{\varepsilon_{k-1}/512 \cdot 3\varepsilon_{k-1}/4}$$

$$\leq \quad \varepsilon_{k-1}/8$$

where the first inequality is from Equation (17) of Invariant 2, the second inequality uses Invariant 1, the third inequality follows from the optimality of $\hat{h}_{k-1}$ and triangle inequality, the fourth inequality uses Equation (21), the fifth inequality uses the fact that $\sqrt{A+B} \leq \sqrt{A} + \sqrt{B}$ and $\sigma_{k-1} \leq \varepsilon_{k-1}/512$, which is from Equation (18) of Invariant 2, the last inequality again utilizes the Equation (18) of Invariant 2. □

**Lemma 6.** *Suppose Invariants 1, 2, and 3 hold in epoch $k-1$ conditioned on event $F_{k-1}$. Then conditioned on event $F_{k-1}$, the implicit confidence set $V_{k-1} = V(\hat{S}_{k-1}, 3\varepsilon_k/2)$ is such that:*
*(1) If $h \in \mathcal{H}$ satisfies $err_D(h) - err_D(h^*) \leq \varepsilon_k$, then $h$ is in $V_{k-1}$.*
*(2) If $h \in \mathcal{H}$ is in $V_{k-1}$, then $err_D(h) - err_D(h^*) \leq \varepsilon_{k-1}$. Hence $V_{k-1} \subseteq B_U(h^*, 2\nu + \varepsilon_{k-1})$.*
*(3) Algorithm 4, **in_disagr_region**, when run on inputs dataset $\hat{S}_{k-1}$, threshold $3\varepsilon_k/2$, unlabeled example $x$, returns 1 if and only if $x$ is in $R_{k-1}$.*

*Proof.* (1) Let $h$ be a classifier with $err_D(h) - err_D(h^*) \leq \varepsilon_k = \varepsilon_{k-1}/2$. Then, by Lemma 4, one has $\mathrm{err}(h, \hat{S}_{k-1}) - \mathrm{err}(\hat{h}_{k-1}, \hat{S}_{k-1}) \leq 3\varepsilon_{k-1}/4 = 3\varepsilon_k/2$. Hence, $h$ is in $V_{k-1}$.
(2) Fix any $h$ in $V_{k-1}$, by definition of $V_{k-1}$,

$$\mathrm{err}(h, \hat{S}_{k-1}) - \mathrm{err}(\hat{h}_{k-1}, \hat{S}_{k-1}) \leq 3\varepsilon_k/2 = 3\varepsilon_{k-1}/4 \tag{22}$$

Recall that from Lemma 5,

$$err_D(\hat{h}_{k-1}) - err_D(h^*) \leq \varepsilon_{k-1}/8$$

Thus for classifier $h$, applying Invariant 1 by taking $h' := \hat{h}_{k-1}$, we get
$$\text{err}(h, S_{k-1}) - \text{err}(\hat{h}_{k-1}, S_{k-1}) \leq \text{err}(h, \hat{S}_{k-1}) - \text{err}(\hat{h}_{k-1}, \hat{S}_{k-1}) + \varepsilon_{k-1}/32 \qquad (23)$$
Therefore,
$$\text{err}_D(h) - \text{err}_D(\hat{h}_{k-1})$$
$$\leq \quad \text{err}(h, S_{k-1}) - \text{err}(\hat{h}_{k-1}, S_{k-1}) + \sigma_{k-1} + \sqrt{\sigma_{k-1}\rho_{S_{k-1}}(h, \hat{h}_{k-1})}$$
$$\leq \quad \text{err}(h, S_{k-1}) - \text{err}(\hat{h}_{k-1}, S_{k-1}) + \sigma_{k-1} + \sqrt{\sigma_{k-1}(\text{err}(h, \hat{S}_{k-1}) + \text{err}(\hat{h}_{k-1}, \hat{S}_{k-1}))}$$
$$\leq \quad \text{err}(h, \hat{S}_{k-1}) - \text{err}(\hat{h}_{k-1}, \hat{S}_{k-1}) + \sigma_{k-1} + \sqrt{\sigma_{k-1}(\text{err}(h, \hat{S}_{k-1}) + \text{err}(\hat{h}_{k-1}, \hat{S}_{k-1}))} + \varepsilon_{k-1}/16$$
$$\leq \quad 13\varepsilon_{k-1}/16 + \sigma_{k-1} + \sqrt{\sigma_{k-1}(2\text{err}(\hat{h}_{k-1}, \hat{S}_{k-1}) + 3\varepsilon_{k-1}/4)}$$
$$\leq \quad 13\varepsilon_{k-1}/16 + \sigma_{k-1} + \sqrt{2\sigma_{k-1}\text{err}(\hat{h}_{k-1}, \hat{S}_{k-1})} + \sqrt{\varepsilon_{k-1}/512 \cdot 3\varepsilon_{k-1}/4}$$
$$\leq \quad 7\varepsilon_{k-1}/8$$
where the first inequality is from Equation (17) of Invariant 2, the second inequality uses the fact that $\rho_{\hat{S}_{k-1}}(h, h') = \rho_{S_{k-1}}(h, h') \leq \text{err}(h, \hat{S}_{k-1}) + \text{err}(h', \hat{S}_{k-1})$ for $h, h' \in \mathcal{H}$, the third inequality uses Equation (23); the fourth inequality is from Equation (22); the fifth inequality is from the fact that $\sqrt{A + B} \leq \sqrt{A} + \sqrt{B}$ and $\sigma_{k-1} \leq \varepsilon_{k-1}/512$, which is from Equation (18) of Invariant 2, the last inequality again follows from Equation (18) of Invariant 2 and algebra.

In conjunction with the fact that $\text{err}_D(\hat{h}_{k-1}) - \text{err}_D(h^*) \leq \varepsilon_{k-1}/8$, this implies
$$\text{err}_D(h) - \text{err}_D(h^*) \leq \varepsilon_{k-1}$$
By triangle inequality, $\rho(h, h^*) \leq 2\nu + \varepsilon_{k-1}$, hence $h \in B_U(h^*, 2\nu + \varepsilon_{k-1})$. In summary $V_{k-1} \subseteq B_U(h^*, 2\nu + \varepsilon_{k-1})$.

(3) Follows directly from Lemma 3 and the fact that $R_{k-1} = \text{DIS}(V_{k-1})$. $\qquad\qquad \square$

## B.2 Training the Difference Classifier

Recall that $\Delta(r) = \text{DIS}(B_U(h^*, r))$ is the disagreement region of the disagreement ball centered around $h^*$ with radius $r$.

**Lemma 7** (Difference Classifier Invariant). *There is a numerical constant $c_1 > 0$ such that the following holds. Suppose that Invariants 1 and 2 hold at the end of epoch $k - 1$ conditioned on event $F_{k-1}$ and that Algorithm 2 has inputs unlabeled data distribution $U$, oracle $O$, $\varepsilon = \varepsilon_k/128$, hypothesis class $\mathcal{H}^{df}$, $\delta = \delta_k/2$, previous labeled dataset $\hat{S}_{k-1}$. Then conditioned on event $F_k$,*
*(1) $\hat{h}_k^{df}$, the output of Algorithm 2, maintains Invariant 3.*
*(2)(Label Complexity: Part 1.) The number of label queries made to $O$ is at most*
$$m_{k,1} \leq c_1 \left( \frac{\mathbb{P}_U(x \in \Delta(2\nu + \varepsilon_{k-1}))}{\varepsilon_k} \left( d' \ln \frac{1}{\varepsilon_k} + \ln \frac{1}{\delta_k} \right) \right)$$

*Proof.* (1) Recall that $F_k = F_{k-1} \cap E_k^1 \cap E_k^2$, where $E_k^1, E_k^2$ are defined in Subsection A.4. Suppose event $F_k$ happens.

**Proof of Equation** (19). Recall that $\hat{h}_k^{df}$ is the optimal solution of optimization problem (2). We have by feasibility and the fact that on event $E_k^3$, $2\hat{p}_k \geq \mathbb{P}_{\mathscr{D}}(x \in R_{k-1})$,
$$\mathbb{P}_{\mathscr{A}_k'}(\hat{h}_k^{df}(x) = -1, y_O \neq y_W) \leq \frac{\varepsilon_k}{256\hat{p}_k} \leq \frac{\varepsilon_k}{128\mathbb{P}_{\mathscr{D}}(x \in R_{k-1})}$$
By definition of event $E_k^2$, this implies
$$\mathbb{P}_{\mathscr{A}_k}(\hat{h}_k^{df}(x) = -1, y_O \neq y_W)$$
$$\leq \quad \mathbb{P}_{\mathscr{A}_k'}(\hat{h}_k^{df}(x) = -1, y_O \neq y_W) + \sqrt{\mathbb{P}_{\mathscr{A}_k'}(\hat{h}_k^{df}(x) = -1, y_O \neq y_W)\frac{\varepsilon_k}{1024\mathbb{P}_{\mathscr{D}}(x \in R_{k-1})}} + \frac{\varepsilon_k}{1024\mathbb{P}_{\mathscr{D}}(x \in R_{k-1})}$$
$$\leq \quad \frac{\varepsilon_k}{64\mathbb{P}_{\mathscr{D}}(x \in R_{k-1})}$$

Indicating
$$\mathbb{P}_{\mathscr{D}}(\hat{h}_k^{df}(x) = -1, y_O \neq y_W, x \in R_{k-1}) \leq \frac{\varepsilon_k}{64}$$

**Proof of Equation** (20). By definition of $h_k^{df}$ in Subsection A.4, $h_k^{df}$ is such that:
$$\mathbb{P}_{\mathscr{D}}(h_k^{df}(x) = +1, x \in \Delta(2\nu + \varepsilon_{k-1})) \leq \alpha(2\nu + \varepsilon_{k-1}, \varepsilon_k/512)$$
$$\mathbb{P}_{\mathscr{D}}(h_k^{df}(x) = -1, y_O \neq y_W, x \in \Delta(2\nu + \varepsilon_{k-1})) \leq \varepsilon_k/512$$
By item (2) of Lemma 6, we have $R_{k-1} \subseteq \mathrm{DIS}(\mathrm{B}_U(h^*, 2\nu + \varepsilon_{k-1}))$, thus
$$\mathbb{P}_{\mathscr{D}}(h_k^{df}(x) = +1, x \in R_{k-1}) \leq \alpha(2\nu + \varepsilon_{k-1}, \varepsilon_k/512) \tag{24}$$

$$\mathbb{P}_{\mathscr{D}}(h_k^{df}(x) = -1, y_O \neq y_W, x \in R_{k-1}) \leq \varepsilon_k/512 \tag{25}$$
Equation (25) implies that
$$\mathbb{P}_{\mathscr{A}_k}(h_k^{df}(x) = -1, y_O \neq y_W) \leq \frac{\varepsilon_k}{512\mathbb{P}_{\mathscr{D}}(x \in R_{k-1})} \tag{26}$$
Recall that $\mathscr{A}'_k$ is the dataset subsampled from $\mathscr{A}_k$ in line 3 of Algorithm 2. By definition of event $E_k^1$, we have that for $h_k^{df}$,

$$\mathbb{P}_{\mathscr{A}'_k}(h_k^{df}(x) = -1, y_O \neq y_W)$$
$$\leq \mathbb{P}_{\mathscr{A}_k}(h_k^{df}(x) = -1, y_O \neq y_W) + \sqrt{\mathbb{P}_{\mathscr{A}_k}(h_k^{df}(x) = -1, y_O \neq y_W)\frac{\varepsilon_k}{1024\mathbb{P}_{\mathscr{D}}(x \in R_{k-1})}} + \frac{\varepsilon_k}{1024\mathbb{P}_{\mathscr{D}}(x \in R_{k-1})}$$
$$\leq \frac{\varepsilon_k}{256\mathbb{P}_{\mathscr{D}}(x \in R_{k-1})} \leq \frac{\varepsilon_k}{256\hat{p}_k}$$

where the second inequality is from Equation (26), and the last inequality is from the fact that $\hat{p}_k \leq \mathbb{P}_{\mathscr{D}}(x \in R_{k-1})$. Hence, $h_k^{df}$ is a feasible solution to the optimization problem (2). Thus,

$$\mathbb{P}_{\mathscr{A}_k}(\hat{h}_k^{df}(x) = +1)$$
$$\leq \mathbb{P}_{\mathscr{A}'_k}(\hat{h}_k^{df}(x) = +1) + \sqrt{\mathbb{P}_{\mathscr{A}'_k}(\hat{h}_k^{df}(x) = +1)\frac{\varepsilon_k}{1024\mathbb{P}_{\mathscr{D}}(x \in R_{k-1})}} + \frac{\varepsilon_k}{1024\mathbb{P}_{\mathscr{D}}(x \in R_{k-1})}$$
$$\leq 2(\mathbb{P}_{\mathscr{A}'_k}(\hat{h}_k^{df}(x) = +1) + \frac{\varepsilon_k}{1024\mathbb{P}_{\mathscr{D}}(x \in R_{k-1})})$$
$$\leq 2(\mathbb{P}_{\mathscr{A}'_k}(h_k^{df}(x) = +1) + \frac{\varepsilon_k}{1024\mathbb{P}_{\mathscr{D}}(x \in R_{k-1})})$$
$$\leq 2((\mathbb{P}_{\mathscr{A}_k}(h_k^{df}(x) = +1) + \sqrt{\mathbb{P}_{\mathscr{A}_k}(h_k^{df}(x) = +1)\frac{\varepsilon_k}{1024\mathbb{P}_{\mathscr{D}}(x \in R_{k-1})}} + \frac{\varepsilon_k}{1024\mathbb{P}_{\mathscr{D}}(x \in R_{k-1})}) + \frac{\varepsilon_k}{1024\mathbb{P}_{\mathscr{D}}(x \in R_{k-1})})$$
$$\leq 6(\mathbb{P}_{\mathscr{A}_k}(h_k^{df}(x) = +1) + \frac{\varepsilon_k}{1024\mathbb{P}_{\mathscr{D}}(x \in R_{k-1})})$$

where the first inequality is by definition of event $E_k^1$, the second inequality is by algebra, the third inequality is by optimality of $\hat{h}_k^{df}$ in (2), $\mathbb{P}_{\mathscr{A}'_k}(\hat{h}_k^{df}(x) = +1) \leq \mathbb{P}_{\mathscr{A}'_k}(h_k^{df}(x) = +1)$, the fourth inequality is by definition of event $E_k^1$, the fifth inequality is by algebra.

Therefore,
$$\mathbb{P}_{\mathscr{D}}(\hat{h}_k^{df}(x) = +1, x \in R_{k-1}) \leq 6(\mathbb{P}_{\mathscr{D}}(h_k^{df}(x) = +1, x \in R_{k-1}) + \varepsilon_k/1024) \leq 6(\alpha(2\nu + \varepsilon_{k-1}, \varepsilon_k/512) + \varepsilon_k/1024)$$
$$\tag{27}$$
where the second inequality follows from Equation (24). This establishes the correctness of Invariant 3.

(2) The number of label requests to $O$ follows from line 3 of Algorithm 2 (see Equation (16)). That is, we can choose $c_1$ large enough (independently of $k$), such that

$$m_{k,1} \leq c_1\Big(\frac{\mathbb{P}_{\mathscr{D}}(x \in R_{k-1})}{\varepsilon_k}(d'\ln\frac{1}{\varepsilon_k} + \ln\frac{1}{\delta_k})\Big) \leq c_1\Big(\frac{\mathbb{P}_U(x \in \Delta(2\nu + \varepsilon_{k-1}))}{\varepsilon_k}(d'\ln\frac{1}{\varepsilon_k} + \ln\frac{1}{\delta_k})\Big)$$

where in the second step we use the fact that on event $F_k$, by item (2) of Lemma 6, $R_{k-1} \subseteq \mathrm{DIS}(\mathrm{B}_U(h^*, 2\nu + \varepsilon_{k-1}))$, thus $\mathbb{P}_{\mathscr{D}}(x \in R_{k-1}) \leq \mathbb{P}_{\mathscr{D}}(x \in \Delta(2\nu + \varepsilon_{k-1})) = \mathbb{P}_U(x \in \Delta(2\nu + \varepsilon_{k-1}))$. □

## B.3 Adaptive Subsampling

---

**Algorithm 5** Adaptive Active Learning using Difference Classifier

---

1: Input: Unlabeled data distribution $U$, oracles $W$ and $O$, difference classifier $h^{df}$, target excess error $\varepsilon$, confidence $\delta$, previous labeled dataset $\hat{T}$.
2: Output: Parameter $\sigma$, labeled dataset $\hat{S}$.
3: Let $\hat{h} = \text{CONS-LEARN}_{\mathscr{H}}(\emptyset, \hat{T})$.
4: **for** $t = 1, 2, \ldots,$ **do**
5:      Let $\delta^t = \delta/t(t+1)$. Define: $\sigma(2^t, \delta^t) = \frac{8}{2^t}(2d\ln\frac{2e2^t}{d} + \ln\frac{24}{\delta^t})$.
6:      Draw $2^t$ examples from $U$ to form $S^{t,U}$.
7:      **for** each $x \in S^{t,U}$ **do**:
8:          **if in_disagr_region**$(\hat{T}, \frac{3\varepsilon}{2}, x) = 0$ **then**   # $x$ is inside the agreement region
9:              Add $(x, \hat{h}(x))$ to $\hat{S}^t$.
10:          **else** # $x$ is inside the disagreement region
11:              If $h^{df}(x) = +1$, query $O$ for the label $y$ of $x$, otherwise query $W$. Add $(x, y)$ to $\hat{S}^t$.
12:          **end if**
13:      **end for**
14:      Train $\hat{h}^t \leftarrow \text{CONS-LEARN}_{\mathscr{H}}(\emptyset, \hat{S}^t)$.
15:      **if** $\sigma(2^t, \delta^t) + \sqrt{\sigma(2^t, \delta^t)\text{err}(\hat{h}^t, \hat{S}^t)} \leq \varepsilon/512$ **then**
16:          $t_0 \leftarrow t$, **break**
17:      **end if**
18: **end for**
19: **return** $\sigma \leftarrow \sigma(2^{t_0}, \delta^{t_0})$, $\hat{S} \leftarrow \hat{S}^{t_0}$.

---

**Lemma 8.** *There is a numerical constant $c_2 > 0$ such that the following holds. Suppose Invariants 1, 2, and 3 hold in epoch $k-1$ on event $F_{k-1}$; Algorithm 5 receives inputs unlabeled distribution $U$, classifier $\hat{h}_{k-1}$, difference classifier $\hat{h}^{df} = \hat{h}_k^{df}$, target excess error $\varepsilon = \varepsilon_k$, confidence $\delta = \delta_k/2$, previous labeled dataset $\hat{S}_{k-1}$. Then on event $F_k$,*
*(1) $\hat{S}_k$, the output of Algorithm 5, maintains Invariants 1 and 2.*
*(2) (Label Complexity: Part 2.) The number of label queries to $O$ in Algorithm 5 is at most:*

$$m_{k,2} \leq c_2\left(\frac{(\nu + \varepsilon_k)(\alpha(2\nu + \varepsilon_{k-1}, \varepsilon_k/512) + \varepsilon_k)}{\varepsilon_k^2} \cdot d\left(\ln^2\frac{1}{\varepsilon_k} + \ln^2\frac{1}{\delta_k}\right)\right)$$

*Proof.* (1) Recall that $F_k = F_{k-1} \cap E_k^1 \cap E_k^2$, where $E_k^1$, $E_k^2$ are defined in Subsection A.4. Suppose event $F_k$ happens.

**Proof of Invariant 1.**   We consider a pair of classifiers $h, h' \in \mathscr{H}$, where $h$ is an arbitrary classifier in $\mathscr{H}$ and $h'$ has excess error at most $\varepsilon_k$.

At iteration $t = t_0(k)$ of Algorithm 5, the breaking criteron in line 14 is met, i.e.

$$\sigma(2^{t_0(k)}, \delta_k^{t_0(k)}) + \sqrt{\sigma(2^{t_0(k)}, \delta_k^{t_0(k)})\text{err}(\hat{h}^{t_0(k)}, \hat{S}_k^{t_0(k)})} \leq \varepsilon_k/512 \qquad (28)$$

First we expand the definition of $\text{err}(h, S_k)$ and $\text{err}(h, \hat{S}_k)$ respectively:

$$\text{err}(h, S_k) = \mathbb{P}_{\mathscr{S}_k}(\hat{h}_k^{df}(x) = +1, h(x) \neq y_O, x \in R_{k-1}) + \mathbb{P}_{\mathscr{S}_k}(\hat{h}_k^{df}(x) = -1, h(x) \neq y_O, x \in R_{k-1}) + \mathbb{P}_{\mathscr{S}_k}(h(x) \neq y_O, x \notin R_{k-1})$$

$$\text{err}(h, \hat{S}_k) = \mathbb{P}_{\mathscr{S}_k}(\hat{h}_k^{df}(x) = +1, h(x) \neq y_O, x \in R_{k-1}) + \mathbb{P}_{\mathscr{S}_k}(\hat{h}_k^{df}(x) = -1, h(x) \neq y_W, x \in R_{k-1}) + \mathbb{P}_{\mathscr{S}_k}(h(x) \neq h^*(x), x \notin R_{k-1})$$

where we use the fact that by Lemma 6, for all examples $x \notin R_{k-1}$, $\hat{h}_{k-1}(x) = h^*(x)$.

We next show that $\mathbb{P}_{\mathscr{S}_k}(\hat{h}_k^{df}(x) = -1, h(x) \neq y_O, x \in R_{k-1})$ is close to $\mathbb{P}_{\mathscr{S}_k}(\hat{h}_k^{df}(x) = -1, h(x) \neq y_W, x \in R_{k-1})$.

From Lemma 7, we know that conditioned on event $F_k$,

$$\mathbb{P}_{\mathscr{D}}(\hat{h}_k^{df}(x) = -1, y_O \neq y_W, x \in R_{k-1}) \leq \varepsilon_k/64$$

In the meantime, from Equation (28), $\gamma(2^{t_0(k)}, \delta_k^{t_0(k)}) \leq \sigma(2^{t_0(k)}, \delta_k^{t_0(k)}) \leq \varepsilon_k/512$. Recall that $\mathscr{S}_k = \mathscr{S}_k^{t_0(k)}$. Therefore, by definition of $E_k^2$,

$$\mathbb{P}_{\mathscr{S}_k}(\hat{h}_k^{df}(x) = -1, y_O \neq y_W, x \in R_{k-1})$$

$$\leq \quad \mathbb{P}_{\mathscr{D}}(\hat{h}_k^{df}(x) = -1, y_O \neq y_W, x \in R_{k-1}) + \sqrt{\mathbb{P}_{\mathscr{D}}(\hat{h}_k^{df}(x) = -1, y_O \neq y_W, x \in R_{k-1})\gamma(2^{t_0(k)}, \delta_k^{t_0(k)})} + \gamma(2^{t_0(k)}, \delta_k^{t_0(k)})$$

$$\leq \quad \mathbb{P}_{\mathscr{D}}(\hat{h}_k^{df}(x) = -1, y_O \neq y_W, x \in R_{k-1}) + \sqrt{\mathbb{P}_{\mathscr{D}}(\hat{h}_k^{df}(x) = -1, y_O \neq y_W, x \in R_{k-1})\varepsilon_k/512} + \varepsilon_k/512$$

$$\leq \quad \varepsilon_k/32$$

By triangle inequality, for all classifier $h_0 \in \mathscr{H}$,

$$|\mathbb{P}_{\mathscr{S}_k}(\hat{h}_k^{df}(x) = -1, h_0(x) \neq y_O, x \in R_{k-1}) - \mathbb{P}_{\mathscr{S}_k}(\hat{h}_k^{df}(x) = -1, h_0(x) \neq y_W, x \in R_{k-1})| \leq \varepsilon_k/32 \quad (29)$$

Specifically for $h$ and $h'$, Equation (29) hold:

$$|\mathbb{P}_{\mathscr{S}_k}(\hat{h}_k^{df}(x) = -1, h(x) \neq y_O, x \in R_{k-1}) - \mathbb{P}_{\mathscr{S}_k}(\hat{h}_k^{df}(x) = -1, h(x) \neq y_W, x \in R_{k-1})| \leq \varepsilon_k/32$$

$$|\mathbb{P}_{\mathscr{S}_k}(\hat{h}_k^{df}(x) = -1, h'(x) \neq y_O, x \in R_{k-1}) - \mathbb{P}_{\mathscr{S}_k}(\hat{h}_k^{df}(x) = -1, h'(x) \neq y_W, x \in R_{k-1})| \leq \varepsilon_k/32$$

Combining, we get:

$$(\mathbb{P}_{\mathscr{S}_k}(\hat{h}_k^{df}(x) = -1, h(x) \neq y_W, x \in R_{k-1}) - \mathbb{P}_{\mathscr{S}_k}(\hat{h}_k^{df}(x) = -1, h'(x) \neq y_W, x \in R_{k-1})) \quad (30)$$

$$- \quad (\mathbb{P}_{\mathscr{S}_k}(\hat{h}_k^{df}(x) = -1, h(x) \neq y_O, x \in R_{k-1}) - \mathbb{P}_{\mathscr{S}_k}(\hat{h}_k^{df}(x) = -1, h'(x) \neq y_O, x \in R_{k-1})) \leq \varepsilon_k/16$$

We now show the labels inferred in the region $\mathscr{X} \setminus R_{k-1}$ is "favorable" to the classifiers whose excess error is at most $\varepsilon_k/2$.
By triangle inequality,

$$\mathbb{P}_{\mathscr{S}_k}(h(x) \neq y_O, x \notin R_{k-1}) - \mathbb{P}_{\mathscr{S}_k}(h^*(x) \neq y_O, x \notin R_{k-1}) \leq \mathbb{P}_{\mathscr{S}_k}(h(x) \neq h^*(x), x \notin R_{k-1})$$

By Lemma 6, since $h'$ has excess error at most $\varepsilon_k$, $h'$ agrees with $h^*$ on all $x$ inside $\mathscr{X} \setminus R_{k-1}$ on event $F_{k-1}$, hence $\mathbb{P}_{\mathscr{S}_k}(h'(x) \neq h^*(x), x \notin R_{k-1}) = 0$. This gives

$$\mathbb{P}_{\mathscr{S}_k}(h(x) \neq y_O, x \notin R_{k-1}) - \mathbb{P}_{\mathscr{S}_k}(h'(x) \neq y_O, x \notin R_{k-1})$$

$$\leq \quad \mathbb{P}_{\mathscr{S}_k}(h(x) \neq h^*(x), x \notin R_{k-1}) - \mathbb{P}_{\mathscr{S}_k}(h'(x) \neq h^*(x), x \notin R_{k-1}) \quad (31)$$

Combining Equations (30) and (31), we conclude that

$$\mathrm{err}(h, S_k) - \mathrm{err}(h', S_k) \leq \mathrm{err}(h, \hat{S}_k) - \mathrm{err}(h', \hat{S}_k) + \varepsilon_k/16$$

This establishes the correctness of Invariant 1.

**Proof of Invariant 2.** Recall by definition of $E_k^2$ the following concentration results hold for all $t \in \mathbb{N}$:

$$|(\mathrm{err}(h, S_k^t) - \mathrm{err}(h', S_k^t)) - (\mathrm{err}_D(h) - \mathrm{err}_D(h'))| \leq \sigma(2^t, \delta_k^t) + \sqrt{\sigma(2^t, \delta_k^t)\rho_{S_k^t}(h, h')}$$

In particular, for iteration $t_0(k)$ we have

$$|(\mathrm{err}(h, S_k^{t_0(k)}) - \mathrm{err}(h', S_k^{t_0(k)})) - (\mathrm{err}_D(h) - \mathrm{err}_D(h'))| \leq \sigma(2^{t_0(k)}, \delta_k^{t_0(k)}) + \sqrt{\sigma(2^{t_0(k)}, \delta_k^{t_0(k)})\rho_{S_k^{t_0(k)}}(h, h')}$$

Recall that $\hat{S}_k = \hat{S}_k^{t_0(k)}$, $\hat{h}_k = \hat{h}_k^{t_0(k)}$, and $\sigma_k = \sigma(2^{t_0(k)}, \delta_k^{t_0(k)})$, hence the above is equivalent to

$$|(\mathrm{err}(h, S_k) - \mathrm{err}(h', S_k)) - (\mathrm{err}_D(h) - \mathrm{err}_D(h'))| \leq \sigma_k + \sqrt{\sigma_k \rho_{S_k}(h, h')} \quad (32)$$

Equation (32) establishes the correctness of Equation (17) of Invariant 2. Equation (18) of Invariant 2 follows from Equation (28).

(2) We define $\tilde{h}_k = \operatorname{argmin}_{h \in \mathcal{H}} \operatorname{err}_{\hat{D}_k}(h)$, and define $\tilde{v}_k$ to be $\operatorname{err}_{\hat{D}_k}(\tilde{h}_k)$. To prove the bound on the number of label requests, we first claim that if $t$ is sufficiently large that

$$\sigma(2^t, \delta_k^t) + \sqrt{\sigma(2^t, \delta_k^t)\tilde{v}_k} \leq \varepsilon_k/1536 \tag{33}$$

then the algorithm will satisfy the breaking criterion at line 14 of Algorithm 5, that is, for this value of $t$,

$$\sigma(2^t, \delta_k^t) + \sqrt{\sigma(2^t, \delta_k^t)\operatorname{err}(\hat{h}^t, \hat{S}_k^t)} \leq \varepsilon_k/512 \tag{34}$$

Indeed, by definition of $E_k^2$, if event $F_k$ happens,

$$\operatorname{err}(\tilde{h}_k, \hat{S}_k^t)$$
$$\leq \quad \operatorname{err}_{\hat{D}_k}(\tilde{h}_k) + \sigma(2^t, \delta_k^t) + \sqrt{\sigma(2^t, \delta_k^t)\operatorname{err}_{\hat{D}_k}(\tilde{h}_k)}$$
$$= \quad \tilde{v}_k + \sigma(2^t, \delta_k^t) + \sqrt{\sigma(2^t, \delta_k^t)\tilde{v}_k} \tag{35}$$

Therefore,

$$\sigma(2^t, \delta_k^t) + \sqrt{\sigma(2^t, \delta_k^t)\operatorname{err}(\hat{h}_k^t, \hat{S}_k^t)}$$
$$\leq \quad \sigma(2^t, \delta_k^t) + \sqrt{\sigma(2^t, \delta_k^t)\operatorname{err}(\tilde{h}_k, \hat{S}_k^t)}$$
$$\leq \quad \sigma(2^t, \delta_k^t) + \sqrt{\sigma(2^t, \delta_k^t)(2\tilde{v}_k + 2\sigma(2^t, \delta_k^t))}$$
$$\leq \quad 3\sigma(2^t, \delta_k^t) + 2\sqrt{\sigma(2^t, \delta_k^t)\tilde{v}_k}$$
$$\leq \quad \varepsilon_k/512$$

where the first inequality is from the optimality of $\hat{h}_k^t$, the second inequality is from Equation (35), the third inequality is by algebra, the last inequality follows from Equation (33). The claim follows. Next, we solve for the minimum $t$ that satisfies (33), which is an upper bound of $t_0(k)$. Fact 1 implies that there is a numerical constant $c_3 > 0$ such that

$$2^{t_0(k)} \leq c_3 \frac{\tilde{v}_k + \varepsilon_k}{\varepsilon_k^2} (d \ln \frac{1}{\varepsilon_k} + \ln \frac{1}{\delta_k}))$$

Thus, there is a numerical constant $c_4 > 0$ such that

$$t_0(k) \leq c_4 (\ln d + \ln \frac{1}{\varepsilon_k} + \ln \ln \frac{1}{\delta_k})$$

Hence, there is a numerical constant $c_5 > 0$ (that does not depend on $k$) such that the following holds. If event $F_k$ happens, then the number of label queries made by Algorithm 5 to $O$ can be bounded as follows:

$$
\begin{aligned}
m_{k,2} &= \sum_{t=1}^{t_0(k)} |S_k^{t,U} \cap \{x : \hat{h}_k^{df}(x) = +1\} \cap R_{k-1}| \\
&= \sum_{t=1}^{t_0(k)} 2^t \mathbb{P}_{\mathscr{S}_k^t}(\hat{h}_k^{df}(x) = +1, x \in R_{k-1}) \\
&\leq \sum_{t=1}^{t_0(k)} 2^t (2\mathbb{P}_{\mathscr{D}}(\hat{h}_k^{df}(x) = +1, x \in R_{k-1}) + 2 \cdot 4 \frac{\ln \frac{2}{\delta_k^t}}{2^t}) \\
&\leq 4 \cdot 2^{t_0(k)} \mathbb{P}_{\mathscr{D}}(\hat{h}_k^{df}(x) = +1, x \in R_{k-1}) + 8 \cdot t_0(k) \ln \frac{2}{\delta_k^{t_0(k)}} \\
&\leq c_5 \left( (\frac{(\tilde{v}_k + \varepsilon_k)\mathbb{P}_{\mathscr{D}}(\hat{h}_k^{df}(x) = +1, x \in R_{k-1})}{\varepsilon_k^2} + 1) \cdot d(\ln^2 \frac{1}{\varepsilon_k} + \ln^2 \frac{1}{\delta_k})) \right) \\
&\leq c_5 \left( (\frac{(\tilde{v}_k + \varepsilon_k) \cdot 6(\alpha(2v + \varepsilon_{k-1}, \varepsilon_k/512) + \varepsilon_k/1024)}{\varepsilon_k^2} + 1) \cdot d(\ln^2 \frac{1}{\varepsilon_k} + \ln^2 \frac{1}{\delta_k})) \right)
\end{aligned}
$$

where the second equality is from the fact that $|S_k^{t,U} \cap \{x : \hat{h}_k^{df}(x) = -1\} \cap R_{k-1}| = |S_k^{t,U}| \cdot \mathbb{P}_{\mathscr{S}_k^t}(\hat{h}_k^{df}(x) = -1, x \in R_{k-1})$, in conjunction with $|S_k^{t,U}| = 2^t$; the first inequality is by definition of $E_k^2$, the second and third inequality is from algebra that $t_0(k) \ln \frac{1}{\delta_k^{t_0(k)}} \le c_5 d(\ln^2 \frac{1}{\varepsilon_k} + \ln^2 \frac{1}{\delta_k})$ for some constant $c_5 > 0$, along with the choice of $c_2$, the fourth step is from Lemma 7 which states that Invariant 3 holds at epoch $k$.

What remains to be argued is an upper bound on $\tilde{\nu}_k$. Note that

$$\tilde{\nu}_k$$
$$= \min_{h \in \mathscr{H}} [\mathbb{P}_{\mathscr{D}}(\hat{h}_k^{df}(x) = -1, h(x) \ne y_W, x \in R_{k-1}) + \mathbb{P}_{\mathscr{D}}(\hat{h}_k^{df}(x) = +1, h(x) \ne y_O, x \in R_{k-1}) + \mathbb{P}_{\mathscr{D}}(h(x) \ne h^*(x), x \notin R_{k-1})]$$
$$\le \mathbb{P}_{\mathscr{D}}(\hat{h}_k^{df}(x) = -1, h^*(x) \ne y_W, x \in R_{k-1}) + \mathbb{P}_{\mathscr{D}}(\hat{h}_k^{df}(x) = +1, h^*(x) \ne y_O, x \in R_{k-1})$$
$$\le \mathbb{P}_{\mathscr{D}}(\hat{h}_k^{df}(x) = -1, h^*(x) \ne y_O, x \in R_{k-1}) + \mathbb{P}_{\mathscr{D}}(\hat{h}_k^{df}(x) = +1, h^*(x) \ne y_O, x \in R_{k-1}) + \varepsilon_k/64$$
$$\le \mathbb{P}_{\mathscr{D}}(\hat{h}_k^{df}(x) = -1, h^*(x) \ne y_O, x \in R_{k-1}) + \mathbb{P}_{\mathscr{D}}(\hat{h}_k^{df}(x) = +1, h^*(x) \ne y_O, x \in R_{k-1}) + \mathbb{P}_{\mathscr{D}}(h(x) \ne y_O, x \notin R_{k-1}) + \varepsilon_k/64$$
$$= \nu + \varepsilon_k/64$$

where the first step is by definition of $\text{err}_{\hat{D}_k}(h)$, the second step is by the suboptimality of $h^*$, the third step is by Equation (29), the fourth step is by adding a positive term $\mathbb{P}_{\mathscr{D}}(h(x) \ne y_O, x \notin R_{k-1})$, the fifth step is by definition of $\text{err}_D(h)$. Therefore, we conclude that there is a numerical constant $c_2 > 0$, such that $m_{k,2}$, the number of label requests to $O$ in Algorithm 5 is at most

$$c_2 \left( \frac{(\nu + \varepsilon_k)(\alpha(2\nu + \varepsilon_{k-1}, \varepsilon_k/512) + \varepsilon_k)}{\varepsilon_k^2} \cdot d(\ln^2 \frac{1}{\varepsilon_k} + \ln^2 \frac{1}{\delta_k}) \right)$$

$\square$

## B.4 Putting It Together – Consistency and Label Complexity

*Proof of Lemma 2.* With foresight, pick $c_0 > 0$ to be a large enough constant. We prove the result by induction.

**Base case.** Consider $k = 0$. Recall that $F_0$ is defined as

$$F_0 = \left\{ \text{for all } h, h' \in \mathscr{H}, |(\text{err}(h, S_0) - \text{err}(h', S_0)) - (\text{err}_D(h) - \text{err}_D(h'))| \le \sigma(n_0, \delta_0) + \sqrt{\sigma(n_0, \delta_0)\rho_{S_0}(h, h')} \right\}$$

Note that by definition in Subsection A.3, $\hat{S}_0 = S_0$. Therefore Invariant 1 trivially holds. When $F_0$ happens, Equation (17) of Invariant 2 holds, and $n_0$ is such that $\sqrt{\sigma_0} \le \varepsilon_0/1024$, thus,

$$\sigma_0 + \sqrt{\sigma_0 \text{err}(\hat{h}_0, \hat{S}_0)} \le \varepsilon_0/512$$

which establishes the validity of Equation (18) of Invariant 2.

Meanwhile, the number of label requests to $O$ is

$$n_0 = 64 \cdot 1024^2 (d \ln(512 \cdot 1024^2) + \ln \frac{96}{\delta})) \le c_0(d + \ln \frac{1}{\delta})$$

**Inductive case.** Suppose the claim holds for $k' < k$. The inductive hypothesis states that Invariants 1,2,3 hold in epoch $k - 1$ on event $F_{k-1}$. By Lemma 7 and Lemma 8, Invariants 1,2,3 holds in epoch $k$ on event $F_k$. Suppose $F_k$ happens. By Lemma 7, there is a numerical constant $c_1 > 0$ such that the number of label queries in Algorithm 2 in line 12 is at most

$$m_{k,1} \le c_1 \left( \frac{\mathbb{P}_U(x \in \Delta(2\nu + \varepsilon_{k-1}))}{\varepsilon_k} (d' \ln \frac{1}{\varepsilon_k} + \ln \frac{1}{\delta_k}) \right)$$

Meanwhile, by Lemma 8, there is a numerical constant $c_2 > 0$ such that the number of label queries in Algorithm 5 in line 14 is at most

$$m_{k,2} \le c_2 \left( \frac{(\alpha(2\nu + \varepsilon_{k-1}, \varepsilon_k/512) + \varepsilon_k)(\nu + \varepsilon_k)}{\varepsilon_k^2} \cdot d(\ln^2 \frac{1}{\varepsilon_k} + \ln^2 \frac{1}{\delta_k}) \right)$$

Thus, the number of label requests in total at epoch $k$ is at most

$$
\begin{aligned}
m_k &= m_{k,1} + m_{k,2} \\
&\leq c_0 \Big( \big( \frac{\alpha(2\nu + \varepsilon_{k-1}, \varepsilon_k/512) + \varepsilon_k)(\nu + \varepsilon_k)}{\varepsilon_k^2} d(\ln^2 \frac{1}{\varepsilon_k} + \ln^2 \frac{1}{\delta_k}) + \frac{\mathbb{P}_U(x \in \Delta(2\nu + \varepsilon_{k-1}))}{\varepsilon_k}(d' \ln \frac{1}{\varepsilon_k} + \ln \frac{1}{\delta_k}) \big) \Big)
\end{aligned}
$$

This completes the induction. $\qquad\square$

**Theorem 3** (Consistency). *If $F_{k_0}$ happens, then the classifier $\hat{h}$ returned by Algorithm 1 is such that*

$$
\mathrm{err}_D(\hat{h}) - \mathrm{err}_D(h^*) \leq \varepsilon
$$

*Proof.* By Lemma 2, Invariants 1, 2, 3 hold at epoch $k_0$. Thus by Lemma 5,

$$
\mathrm{err}_D(\hat{h}) - \mathrm{err}_D(h^*) = \mathrm{err}_D(\hat{h}_{k_0}) - \mathrm{err}_D(h^*) \leq \varepsilon_{k_0}/8 \leq \varepsilon
$$

$\qquad\square$

*Proof of Theorem 1.* This is an immediate consequence of Theorem 3. $\qquad\square$

**Theorem 4** (Label Complexity). *If $F_{k_0}$ happens, then the number of label queries made by Algorithm 1 to $O$ is at most*

$$
\tilde{O}\big((\sup_{r \geq \varepsilon} \frac{\alpha(2\nu + r, r/1024)}{2\nu + r}) d(\frac{\nu^2}{\varepsilon^2} + 1) + (\sup_{r \geq \varepsilon} \frac{\mathbb{P}_U(x \in \Delta(2\nu + r))}{2\nu + r}) d'(\frac{\nu}{\varepsilon} + 1)\big)
$$

*Proof.* Conditioned on event $F_{k_0}$, we bound the sum $\sum_{k=0}^{k_0} m_k$.

$$
\begin{aligned}
&\sum_{k=0}^{k_0} m_k \\
\leq\ & c_0(d + \ln \frac{1}{\delta}) + c_0 \Big( \sum_{k=1}^{k_0} \frac{(\alpha(2\nu + \varepsilon_{k-1}, \varepsilon_k/512) + \varepsilon_k)(\nu + \varepsilon_k)}{\varepsilon_k^2} d(\ln^2 \frac{1}{\varepsilon_k} + \ln^2 \frac{1}{\delta_k}) + \frac{\mathbb{P}_U(x \in \Delta(2\nu + \varepsilon_{k-1}))}{\varepsilon_k}(d' \ln \frac{1}{\varepsilon_k} + \ln \frac{1}{\delta_k}) \Big) \\
\leq\ & c_0(d + \ln \frac{1}{\delta}) + c_0 \Big( \sum_{k=1}^{k_0} \frac{(\alpha(2\nu + \varepsilon_{k-1}, \varepsilon_k/512) + \varepsilon_k)(\nu + \varepsilon_k)}{\varepsilon_k^2} d(3\ln^2 \frac{1}{\varepsilon} + 2\ln^2 \frac{1}{\delta}) + \frac{\mathbb{P}_U(x \in \Delta(2\nu + \varepsilon_{k-1}))}{\varepsilon_k}(2d' \ln \frac{1}{\varepsilon} + \ln \frac{1}{\delta}) \Big) \\
\leq\ & (\sup_{r \geq \varepsilon} \frac{\alpha(2\nu + r, r/1024) + r}{2\nu + r}) d(3\ln^2 \frac{1}{\varepsilon} + 2\ln^2 \frac{1}{\delta}) \sum_{k=0}^{k_0} \frac{(\nu + \varepsilon_k)^2}{\varepsilon_k^2} + \sup_{r \geq \varepsilon} \frac{\mathbb{P}_U(x \in \Delta(2\nu + r))}{2\nu + r}(2d' \ln \frac{1}{\varepsilon} + \ln \frac{1}{\delta}) \sum_{k=0}^{k_0} \frac{(\nu + \varepsilon_k)}{\varepsilon_k} \\
\leq\ & \tilde{O}\big((\sup_{r \geq \varepsilon} \frac{\alpha(2\nu + r, r/1024) + r}{2\nu + r}) d(\frac{\nu^2}{\varepsilon^2} + 1) + (\sup_{r \geq \varepsilon} \frac{\mathbb{P}_U(x \in \Delta(2\nu + r))}{2\nu + r}) d'(\frac{\nu}{\varepsilon} + 1)\big)
\end{aligned}
$$

where the first inequality is by Lemma 2, the second inequality is by noticing for all $k \geq 1$, $\ln^2 \frac{1}{\varepsilon_k} + \ln^2 \frac{1}{\delta_k} \leq 3\ln^2 \frac{1}{\varepsilon} + 2\ln^2 \frac{1}{\delta}$ and $d' \ln \frac{1}{\varepsilon_k} + \ln \frac{1}{\delta_k} \leq 2d' \ln \frac{1}{\varepsilon} + \ln \frac{1}{\delta}$, the rest of the derivations follows from standard algebra.

$\qquad\square$

*Proof of Theorem 2.* Item 1 is an immediate consequence of Lemma 2, whereas item 2 is a consequence of Theorem 4. $\qquad\square$

# C  Case Study: Linear Classfication under Uniform Distribution over Unit Ball

We remind the reader the setting of our example in Section 4. $\mathscr{H}$ is the class of homogeneous linear separators on the $d$-dimensional unit ball and $\mathscr{H}^{df}$ is defined to be $\{h\Delta h' : h, h' \in \mathscr{H}\}$. Note that $d'$ is at most $5d$. Furthermore, $U$ is the uniform distribution over the unit ball. $O$ is a deterministic labeler such that $\text{err}_D(h^*) = v > 0$, $W$ is such that there exists a difference classifier $\bar{h}^{df}$ with false negative error 0 for which $\Pr_U(\bar{h}^{df}(x) = 1) \leq g = o(\sqrt{d}v)$. We prove the label complexity bound provided by Corollary 1.

*Proof of Corollary 1.* We claim that under the assumptions of Corollary 1, $\alpha(2v + r, r/1024)$ is at most $g$. Indeed, by taking $h^{df} = \bar{h}^{df}$, observe that

$$P(\bar{h}^{df}(x) = -1, y_W \neq y_O, x \in \Delta(2v + r)) \leq P(\bar{h}^{df}(x) = -1, y_W \neq y_O) = 0$$

$$P(\bar{h}^{df}(x) = +1, x \in \Delta(2v + r)) \leq g$$

This shows that $\alpha(2v + r, 0) \leq g$. Hence, $\alpha(2v + r, r/1024) \leq \alpha(2v + r, 0) \leq g$. Therefore,

$$\sup_{r:r\geq\varepsilon} \frac{\alpha(2v + r, r/1024) + r}{2v + r} \leq \sup_{r\geq\varepsilon} \frac{g + r}{v + r} \leq \max(\frac{g}{v}, 1)$$

Recall that the disagreement coefficient $\theta(2v + r) \leq \sqrt{d}$ for all $r$, and $d' \leq 5d$. Thus, by Theorem 2, the number of label queries to $O$ is at most

$$\tilde{O}\left(d\max(\frac{g}{v}, 1)(\frac{v^2}{\varepsilon^2} + 1) + d^{3/2}\left(1 + \frac{v}{\varepsilon}\right)\right)$$

$\square$

# D  Performance Guarantees for Learning with Respect to Data labeled by $O$ and $W$

An interesting variant of our model is to consider learning from data labeled by a mixture of $O$ and $W$.

Let $D_W$ be the distribution over labeled examples determined by $U$ and $W$, specifically, $\mathbb{P}_{D_W}(x, y) = \mathbb{P}_U(x)\mathbb{P}_W(y|x)$. Let $D'$ be a mixture of $D$ and $D_W$, specifically $D' = (1 - \beta)D + \beta D_W$, for some parameter $\beta > 0$. Define $h'$ to be the best classifier with respect to $D'$, and denote by $v'$ the error of $h'$ with respect to $D'$.

Let $O'$ be the following *mixture oracle*. Given an example $x$, the label $y_{O'}$ is generated as follows. $O'$ flips a coin with bias $\beta$. If it comes up heads, it queries $W$ for the label of $x$ and returns the result; otherwise $O$ is queried and the result returned. It is immediate that the conditional probability induced by $O'$ is $P_{O'}(y|x) = (1 - \beta)\mathbb{P}_O(y|x) + \beta\mathbb{P}_W(y|x)$, and $D'(x, y) = P_{O'}(y|x)P_U(x)$.

**Assumption 2.** *For any $r, \eta > 0$, there exists an $h^{df}_{\eta,r} \in \mathscr{H}^{df}$ with the following properties:*

$$\mathbb{P}_{\mathscr{D}}(h^{df}_{\eta,r}(x) = -1, x \in \Delta(r), y_{O'} \neq y_W) \leq \eta$$

$$\mathbb{P}_{\mathscr{D}}(h^{df}_{\eta,r}(x) = 1, x \in \Delta(r)) \leq \alpha'(r, \eta)$$

Recall that the disagreement coefficient $\theta(r)$ at scale $r$ is $\theta(r) = \sup_{h\in\mathscr{H}}\sup_{r'\geq r} \frac{\mathbb{P}_U(\text{DIS}(B_U(h, r')))}{r'}$, which only depends on the unlabeled data distribution $U$ and does not depend on $\bar{W}$ or $O$.

We have the following corollary.

**Corollary 2** (Learning with respect to Mixture). *Let $d$ be the VC dimension of $\mathscr{H}$ and let $d'$ be the VC dimension of $\mathscr{H}^{df}$. If Assumption 2 holds, and if the error of the best classifier in $\mathscr{H}$ on $D'$ is at most $v'$. Algorithm 1 is run with inputs unlabeled distribution $U$, target excess error $\varepsilon$, confidence $\delta$, labeling oracle $O'$, weak oracle $W$, hypothesis class $\mathscr{H}$, hypothesis class for difference classifier $\mathscr{H}^{df}$, confidence $\delta$. Then with probability $\geq 1 - 2\delta$, the following hold:*

1. *the classifier $\hat{h}$ output by Algorithm 1 satisfies: $err_{D'}(\hat{h}) \leq err_{D'}(h') + \varepsilon$.*

2. *the total number of label queries made by Algorithm 1 to the oracle O is at most:*

$$\tilde{O}\left((1-\beta)\left(\sup_{r \geq \varepsilon} \frac{\alpha'(2v'+r, r/1024) + r}{2v'+r} \cdot d\left(\frac{v'^2}{\varepsilon^2} + 1\right) + \theta(2v'+\varepsilon)d'\left(\frac{v'}{\varepsilon} + 1\right)\right)\right)$$

*Proof Sketch.* Consider running Algorithm 1 in the setting above. By Theorem 1 and Theorem 2, there is an event $F$ such that $\mathbb{P}(F) \geq 1 - \delta$, if event $F$ happens, $\hat{h}$, the classifier learned by Algorithm 1 is such that

$$\mathrm{err}_{D'}(\hat{h}) \leq \mathrm{err}_{D'}(h') + \varepsilon$$

By Theorem 2, the number of label requests to $O'$ is at most

$$m_{O'} = \tilde{O}\left(\sup_{r \geq \varepsilon} \frac{\alpha'(2v'+r, r/1024) + r}{2v'+r} \cdot d\left(\frac{v'^2}{\varepsilon^2} + 1\right) + \theta(2v'+\varepsilon)d'\left(\frac{v'}{\varepsilon} + 1\right)\right)$$

Since $O'$ is simulated by drawing a Bernoulli random variable $Z_i \sim \mathrm{Ber}(1-\beta)$ in each call of $O'$, if $Z_i = 1$, then return $O(x)$, otherwise return $W(x)$. Define event

$$H = \{\sum_{i=1}^{m_{O'}} Z_i \leq 2((1-\beta)m_{O'} + 4\ln\frac{2}{\delta})\}$$

by Chernoff bound, $\mathbb{P}(H) \geq 1 - \delta$. Consider event $J = F \cap H$, by union bound, $\mathbb{P}(J) \geq 1 - 2\delta$. Conditioned on event $J$, the number of label requests to $O$ is at most $\sum_{i=1}^{m_{O'}} Z_i$, which is at most

$$\tilde{O}\left((1-\beta)\left(\sup_{r \geq \varepsilon} \frac{\alpha'(2v'+r, r/1024) + r}{2v'+r} \cdot d\left(\frac{v'^2}{\varepsilon^2} + 1\right) + \theta(2v'+\varepsilon)d'\left(\frac{v'}{\varepsilon} + 1\right)\right)\right)$$

$\square$

# E  Remaining Proofs

*Proof of Fact 2.* (1) First by Lemma 1, $\mathbb{P}_{\mathscr{D}}(x \in R_{k-1})/2 \leq \hat{p}_k \leq \mathbb{P}_{\mathscr{D}}(x \in R_{k-1})$ holds with probability $1 - \delta_k/6$.

Second, for each classifier $h^{df} \in \mathscr{H}^{df}$, define functions $f^1_{h^{df}}$, and $f^2_{h^{df}}$ associated with it. Formally,

$$f^1_{h^{df}}(x, y_O, y_W) = I(h^{df}(x) = -1, y_O \neq y_W)$$

$$f^2_{h^{df}}(x, y_O, y_W) = I(h^{df}(x) = +1)$$

Consider the function class $\mathscr{F}^1 = \{f^1_{h^{df}} : h^{df} \in \mathscr{H}^{df}\}$, $\mathscr{F}^2 = \{f^2_{h^{df}} : h^{df} \in \mathscr{H}^{df}\}$. Note that both $\mathscr{F}^1$ and $\mathscr{F}^2$ have VC dimension $d'$, which is the same as $\mathscr{H}^{df}$. We note that $\mathscr{A}'_k$ is a random sample of size $m_k$ drawn iid from $\mathscr{A}_k$. The fact follows from normalized VC inequality on $\mathscr{F}^1$ and $\mathscr{F}^2$ and the choice of $m_k$ in Algorithm 2 called in epoch $k$, along with union bound. $\square$

*Proof of Fact 3.* For fixed $t$, we note that $S^t_k$ is a random sample of size $2^t$ drawn iid from $D$. By Equation (8), for any fixed $t \in \mathbb{N}$,

$$\mathbb{P}\left(\text{for all } h, h' \in \mathscr{H}, |(\mathrm{err}(h, S^t_k) - \mathrm{err}(h', S^t_k)) - (\mathrm{err}_D(h) - \mathrm{err}_D(h'))| \leq \sigma(2^t, \delta^t_k) + \sqrt{\sigma(2^t, \delta^t_k)\rho_{S^t_k}(h, h')}\right) \geq 1 - \delta^t_k/8$$
(36)

Meanwhile, for fixed $t \in \mathbb{N}$, note that $\hat{S}^t_k$ is a random sample of size $2^t$ drawn iid from $\hat{D}_k$. By Equation (8),

$$\mathbb{P}\left(\text{for all } h, h' \in \mathscr{H}, \mathrm{err}(h, \hat{S}^t_k) - \mathrm{err}_{\hat{D}_k}(h) \leq \sigma(2^t, \delta^t_k) + \sqrt{\sigma(2^t, \delta^t_k)\mathrm{err}_{\hat{D}_k}(h)}\right) \geq 1 - \delta^t_k/8$$ (37)

Moreover, for fixed $t \in \mathbb{N}$, note that $\mathscr{S}^t_k$ is a random sample of size $2^t$ drawn iid from $\mathscr{D}$. By Equation (12),

$$\mathbb{P}\left(\mathbb{P}_{\mathscr{S}^t_k}(\hat{h}^{df}_k(x) = -1, y_O \neq y_W, x \in R_{k-1}) \leq \mathbb{P}_{\mathscr{D}}(\hat{h}^{df}_k(x) = -1, y_O \neq y_W, x \in R_{k-1})\right.$$

$$+ \left. \sqrt{\gamma(2^t, \delta^t_k)\mathbb{P}_{\mathscr{D}}(\hat{h}^{df}_k(x) = -1, y_O \neq y_W, x \in R_{k-1})} + \gamma(2^t, \delta^t_k)\right) \geq 1 - \delta^t_k/8$$ (38)

Finally, for fixed $t \in N$, note that $\mathscr{S}_k^t$ is a random sample of size $2^t$ drawn iid from $\mathscr{D}$. By Equation (12),

$$\mathbb{P}\left(\mathbb{P}_{\mathscr{S}_k^t}(\hat{h}_k^{df}(x) = -1, x \in R_{k-1}) \leq \mathbb{P}_{\mathscr{D}}(\hat{h}_k^{df}(x) = -1, x \in R_{k-1}) + \sqrt{\mathbb{P}_{\mathscr{D}}(\hat{h}_k^{df}(x) = -1, x \in R_{k-1})\gamma(2^t, \delta_k^t)} + \gamma(2^t, \delta_k^t)\right) \geq 1 - \delta_k^t/8$$
(39)

Note that by algebra,

$$\mathbb{P}_{\mathscr{D}}(\hat{h}_k^{df}(x) = -1, x \in R_{k-1}) + \sqrt{\mathbb{P}_{\mathscr{D}}(\hat{h}_k^{df}(x) = -1, x \in R_{k-1})\gamma(2^t, \delta_k^t)} + \gamma(2^t, \delta_k^t) \leq 2(\mathbb{P}_{\mathscr{D}}(\hat{h}_k^{df}(x) = -1, x \in R_{k-1}) + \gamma(2^t, \delta_k^t))$$

Therefore,

$$\mathbb{P}\left(\mathbb{P}_{\mathscr{S}_k^t}(\hat{h}_k^{df}(x) = -1, x \in R_{k-1}) \leq 2(\mathbb{P}_{\mathscr{D}}(\hat{h}_k^{df}(x) = -1, x \in R_{k-1}) + \gamma(2^t, \delta_k^t))\right) \geq 1 - \delta_k^t/12 \quad (40)$$

The proof follows by applying union bound over Equations (36), (37), (38) and (40) and $t \in \mathbb{N}$. $\quad\square$

We emphasize that $\mathscr{S}_k^t$ is chosen iid at random after $\hat{h}_k^{df}$ is determined, thus uniform convergence argument over $\mathscr{H}^{df}$ is not necessary for Equations (38) and (40).

*Proof of Fact 4.* By induction on $k$.

**Base Case.** For $k = 0$, it follows directly from normalized VC inequality that $\mathbb{P}(F_0) \geq 1 - \delta_0$.

**Inductive Case.** Assume $\mathbb{P}(F_{k-1}) \geq 1 - \delta_0 - \ldots - \delta_{k-1}$ holds. By union bound,

$$\mathbb{P}(F_k) \geq \mathbb{P}(F_{k-1} \cap E_k^1 \cap E_k^2) \geq \mathbb{P}(F_{k-1}) - \delta_k/2 - \delta_k/2 \geq \mathbb{P}(F_{k-1}) - \delta_k$$

Hence, $\mathbb{P}(F_k) \geq 1 - \delta_0 - \ldots - \delta_k$. This finishes the induction.
In particular, $\mathbb{P}(F_{k_0}) \geq 1 - \delta_0 - \ldots \delta_{k_0} \geq 1 - \delta$. $\quad\square$