[Reviews · NeurIPS 2015]

Submitted by Assigned_Reviewer_1

1 SUMMARY OF THE PAPER

In this submission, the authors propose to tackle the problem of Active Learning from weak and strong labelers. That is, active learning when the labels can be queried from two different sources. One, the Strong Oracle (noted O in the paper), is costly but reliable while the other, the Weak Oracle (noted W) is marginally cheap but also locally unreliable. The idea behind this is to minimize the use of O by relying on the labels from W as much as possible.

The authors present a new algorithm to learn in this setting. Their solution takes its root on the Disagreement-Based-Active-Learning (DBAL) literature which advocate to only query labels that are in an 'uncertainty region' on the input space. On top of that, the authors propose to learn a 'difference classifier' to determine whether or not O and W return different labels. Hence, for a given datapoint $x$ O is only queried if the two following conditions are met : 1) $x$ is in the disagreement region and 2) the 'difference classifier' predict that W may not return the same label than O.

The main difficulty in this setting seems to come from the fact that the 'difference classifier' might produce false negative (FN) error --that is, when it wrongly predicts agreement between W and O. The proposed solution for this problem is to enforce a maximal FN rate with a cost-sensitive learning procedure for the difference classifier.

The emphasis of the submission is put on keeping the setting as general as possible and this is clearly one of its strong point. The proposed algorithm is extremely general and can be used with any class of classifier. Nevertheless, the authors are able to obtain theoretical guarantees

on the number of query to O. Moreover, those results apply to the very general agnostic PAC setting.

In addition to this, the authors propose a case study for their algorithm in the context of linear classification with uniformly distributed datapoints. They provide extended results for this special case based on their previous theoretical bound on the number of query to O.

Finally, they also quickly discuss the case of learning with data labeled by a mixture of O and W and provide corollary results for this setting.

2 QUALITY

From a scientific point of view, the submission is good and interesting. The generality of the results is remarkable and, to quote the authors, this represent "a step towards a theoretical understanding of active learning from multiple annotators". The proofs presented in the appendices are very thorough and with a great level of detail and every piece of result is fully and properly tackled. Some results are far from trivial and show a very good understanding of the setting and concepts involved in the proofs.

However, I have major concerns about the exposition of this work.

First, the paper seems extremely rushed: there is a plethora of typos, the notations seem also not completely though out, especially in the appendices and the whole dynamic of the paper feels clunky at times. While this clearly impairs the quality of this paper, it is mainly a matter of clarity and therefore I will defer more detailed arguments in the next section. Also, a comprehensive list of typos and style suggestion is given at the end of this review. My second concern is that a fair amount of the interesting stuff is actually in the appendices. A whole subsection of the proposed algorithm is deferred

to them although it is clearly non-trivial, new, and pivotal to the whole learning procedure. Moreover, the submission only contains the major results with no proof. I understand that the proofs are very long and convoluted but a very rough sketch of them would help the reader understand the leading idea. Ironically, this is actually done at the beginning of the major proofs in the appendices. The same remark applies to some intermediate lemmas and theorems that are, in my opinion, core to the comprehension of the submission but only mentioned in the appendices.

Overall the paper is interesting and seems mathematically solid but it feels very rushed and trimmed to fit in the 9 pages limit of the conference. Because of that, and despite its undeniable scientific quality, I do not think that this submission meet the quality standards of the conference. For me it is a typical case of a good and very interesting journal submission that is way too dense, in its present state, for a conference paper.

3 CLARITY

There are several things in this submission that, once added, greatly impair clarity.

1) The various typos and sentences mismatch can be sometimes very confusing, although this can easily be fixed with careful proofreading (see the end of this review for a detailed list).

2) The first subroutine of the proposed algorithm is actually in the appendices. This introduce unintuitive numbering across the different parts of the whole algorithm. In order to unfold it fully, one has to read Alg.1, then Alg.4 (which is in the appendices), followed by Alg.2 and 3. Moreover, it would help if each subroutine was presented as functions. This would vastly improve the readability of the subcalls in Alg.1, especially when it comes to parameters. Finally I think it is would help to have the full algorithm (that is, Alg. 1, 4, 2 and 3) somewhere in one block for easy reference; this may be put in the appendices provided that Alg. 4 is also in the main submission.

3) The notations tend to obfuscate the submission to a point that it is, at least for me, really difficult to follow the reasoning of the authors. In my opinion, it is the whole notation system that should be re-though. In their present state, the notation choices seem somewhat random. This is especially true for classifiers and sampled subsets of examples (in the appendices) for which it is very hard to infer their purpose from their format only.

4) The authors tend to paraphrase themselves at least two times, this is confusing because there is typically very little new information in those parts.

This first happens at the beginning of the section 3 (Algorithm) which is basically is a rewriting of the second part of the introduction, with added notation. Given that the introduction is just 2 pages away, it seems unnecessary to go, again, through a complete explanation of the proposed approach, especially since space seems to be an issue here. The second case is in the beginning of the appendices. The authors reintroduce the notations of the paper. This is very confusing because some old notations are not re-introduced, yet they are used thereafter, while some new notations that could have been used in the main submission are defined, and at least one ($\sigma(n ,\delta)$) is defined for the first time here despite being used liberally in the main submission without any mention of its subsequent definition.

4 ORIGINALITY

The novelty of this paper is clearly the generality of the proposed setting. The authors do a good job on mentioning related works and adequately cite important papers related to their setting. It should be noted that the proposed approach (DBAL coupled with a difference classifier for weak and strong labelers) is extremely close to the one proposed in the work of Malago, Cesa-Bianchi and Renders (cf [14] in the submission's bibliography). The submission feels like an (highly non-trivial) extension of [14] to a far more general setting. While the theoretical results are clearly new and unprecedented, the proposed algorithm is more of an adaptation of the one of [14] to the setting of the submission. Nonetheless, it is my opinion that the theoretical results alone represent a step forward that by itself is worth of a publication. Especially given that the authors also tackle the special cases of linear classification and mixed oracle at the end of the submission.

5. SIGNIFICANCE

This work formally establishes that weak labeler can be used to reduce oracle queries in virtually any instance of active learning with any finite VC-dimension hypothesis class under very mild assumptions. While purely theoretical this is an interesting result that validate the intuitive idea that a strong labeler is not always needed. The assumption that the weak oracle is locally unreliable is very interesting because it matches real case problems where some part of the input space are difficult to classify while others are trivial.

In summary, this submission represent a theoretical step in assessing the usefulness of adding a weak oracle to the regular active learning setting. In this respect, the authors establishes that, it can nearly always be beneficial to do so, provided one use their algorithm. They also quickly discuss the some limit cases where their algorithm provide only marginal benefits over regular DBAL approach.

6. TYPOS & STYLISTIC SUGGESTIONS

Please consider the following changes. While some are just typo correction, others should be taken as style suggestion despite being stated in the imperative for clarity's sake.

l1. You use 'labeler' in your title with only one 'l' but later you use the form with two 'l' in 'labelled'

(l.400). You should pick a form and stick with it for the remaining of the paper. Moreover, since 'labeler' is more related to American english and 'labeller' to British english, you should choose the form that fit best your writing style to stay consistent across your paper.

l17-18: Remove "On the theory side" l121: Replace $\geq$ by "at least" l121: Clarify what "over the sample" means l130: Remove "large" l167: Having "df" as exponent is confusing because $d$ is also the VC-Dim of $H$ l175: Do not use the term 'disagreement' when speaking of the difference between O and W, this is confusing because of BDAL nomenclature

l190: I'm not sure of why $\epsilon_k$ has this value. Is it arbitrary ? l201: Better explain what are $\hat{C}_k$ and $A^U_k$. Right now it is very confusing because one depends on $k$ only but the other depends on $U$ and $k$. Also why the hat on $C$ and not on $A$. I understand this become more clear in the appendices but the reader does not know this at that point. Especially because Alg.4 is not in the paper.

l209-210: Remove "A hard bound is" by "and" and merge with the previous sentence. l215: How $A^U_k$ is related to the disagreement region $DIS(V_t)$ and what is the link between $\phi_k$ and $\phi_t$ l215: the $\bigO$ notation is exactly the same that O the oracle l215: As $\phi_k$ diminishes, does the FN error in $O(\epsilon_k / \phi_k) increase too ? Alg1, l3: What is the use of $\delta_0 = \delta / 4$ ? You could use directly $\delta$ in l.4 because of the $\bigO$ Alg1, l5: $\sigma(.,.)$ is not defined until the appendices. Alg1, l10: You never draw $T^U_k$ prior to its use here. I assume this is a mistake since you explicitly mention drawing $\hat{S}_0$ in l5. Alg1, l10: You do not mention a threshold parameter $\epsilon_{k-1}$ when you introduce Alg.4 (l199-207) Alg1, l15: Place a "EndFor" line above rather than simply rely on indentation Alg1: Change the formatting of the comments or add empty lines to better separate the different subcalls. Right now, the whole algorithms seems messy Alg1: Call Alg 4, 2 and 3 through functions with parameters to improve the general clarity Alg2,l1: U is previously defined as the Unlabeled distribution. Use $A_k^U$ instead. Alg2,l3: It is not clear to me if you re-use the labels queried for $U'$ in the active learning procedure. If you don't, can you do it ? and why ? l263: Does this inclusion hold with probability 1 ? I am under the impression that it should hold with probability $1- \delta$ but it is not clear from the text. l300: Precise what similar means in this context. l315: I cannot find another definition of $\alpha(.,.)$, in that case it means that eq. (3) hold trivialy by definition l315: The probability's distribution is missing in $P(DIS(B_U(h^*, r)))$ l314-317: The whole phrasing here is unclear l336: Replace $\geq$ by "at least" l337: "[...] in epoch $k$ is at most [...]" Eq(5): Order the terms the same way than the DBAL's result, that is: [VC-Dim][\theta(...)][1 + ./.] l353: "firsts" l359: You just stated above that $\sup_{r \geq \epsilon}[...]$ is less than the disagreement coefficient. I get what you mean but you should rephrase this sentence. l375: If you want to use $\bar{}$ here, you should also use $\bar{h}$ instead of $h^*$ l377: The usual result is that, with linear classifier, the VC-dim is $n+1$ where $n$ is the dimensionally of your data. This is clearly not the case here, because you restrict your example to the unit ball. Nonetheless, this can be confusing because your VC-dimension is denoted $d$, which is usually the data dimension (which is also $2$ here). Add a few word to clarify that.

l390: You never introduce the notations $h_w$ l396: See remark on Eq (5) l396: From your previous results, it is expected that $d'$ appears in the second term. You should mention in the main submission that $d' < 5d$ rather than only in the appendices. l407: Why not re-use $h^*$ rather than introduce another new notation ? l409: I don't get the whole $\alpha'$ thing. Is $\alpha$ just a short-hand to $\alpha'(r, \eta)$ ? And what is exactly $\alpha'$ ? l409: Write down the modified version of Alg.1 or at the very least explain in what it is different. l415: I'm not sure if it is supposed to be $O$ or $O'$ here. l421: The conclusion should be in its own section instead of just a paragraph. To some extends this also apply to the late subsections that have the same formatting than the theorems/corollaries/assumptions (inline, bold, with a '.' at the end)

-- Appendices --

l551: Missing Distribution in $B(h^*, r)$ l576: "We begin each epoch $k$ of Algorithm [...]" l579: Do not separate everything with colons in $(x_i, y_{i,O}, y_{i,W})$. This holds for the remaining of the appendices. l579: If you start with $T$, $A$, and $C$ explicitly conditioned on $U$ it should also be the case of $\mathcal{T}$, $\mathcal{A}$ and $\mathcal{C}$ l582: Remind what is $\hat{C}_k$ because you only mention it briefly (l201) l591: Say that Eq(14) is not yet given l593: Are the constants really important ? l597: "to denote" l605: $T^U_k$ vs $T_k$ vs $\mathcal{T}_k$ vs $\hat{T}_k$ is really confusing l608: What if $|T^U_k| < 2^t$ ? l610: I'm not sure of what $S^U_k$ actually is. l613: Does that mean that $S^t_k$ is actually independent of $\mathcal{S}^{t}_k$ ? l616: The way I understand the definition of $\hat{S}_k^t$, the third term is pointless l620: What is $m_{k,2}$ l621: "hatS_k" l622: Is it really necessary, given all the notation already defined, to add one simply to avoid writing $\hat{}$ ? At this point you already have at least defined $S_0$, $\hat{S_0}$, $S_k$, $S_k^t$, $\hat{S}_k^t$, $\mathcal{S}_k^t$, $S_k^{U,t}$. And this is only based on $S$ the same thing is true for $T$, $A$ and $C$.

l626: Do not use them interchangeably, stick with one. l626: "interchangeably" l632&l635: Add a few words on where these constants come from and why it is relevant to explicitly write them. l641: "We will condition on later in the proof" This part seems odd. l684: From my understanding, it seems that it is the distance between the errors of any pair of classifier that will not increase much. I am not sure that it is equivalent to what you write though. l692: "that concentration of empirical [...] to their true errors." The whole phrasing here seems off. l700: I'm not sure that $\hat{h}_k$ is defined. If it is the case, you should recall what it is. l718: Again, this threshold is never mentioned before here and Alg1. l10. l790: Does that mean that $h$ does not make any mistake on $\hat{C}_k$, even in the agnostic case ? l886: What do you mean by "by feasibility" ? l945: "W or O" l948: From my understanding, this holds because $T_k = T_k^U = \mathcal{T}_k$. The right-hand part is only briefly mentioned in a proof above and you should remind it. As for the left-hand part I cannot find it anywhere but I probably missed it.

l955: You have not used $\gamma$ in roughly $400$ lines, remind what it is. l1087: Directly write $F_0$ and "epoch $0$" l1187: "PR_U" I suppose a "\" is missing here l1283: Is the VC-dim always $d'$ or at most $d'$ ? l1288: See l608 l1291 (and below): Replace "for all" with $\forall$
Summary: A good theoretical paper with interesting results. Unfortunately, it is crippled by bad formatting, time constraint and obvious size problem. Either consider serious reworking or submitting your work in full (i.e. with the appendices) at a scientific journal.

Submitted by Assigned_Reviewer_2

The problem is interesting and as the author mentioned this problem has not been previously considered in the literature. To take a step back, what is known about this problem in the passive learning other than the work of Urner, Ben-David and Shamir you cited?

The ideas in the paper are interesting and quite useful but how easy it is to learn the difference classifier? More concretely what is the relationship between d' and d? As authors mentioned, d' can be greater than d but is there an upper bound on d' (in terms of d?)

Assumption 1 seems very strong, btw, specially (3).

The paper can benefit from more clear writing and the algorithms can be moved to appendix to make some more space.

Minor comment: At the end of page 4, the authors mention that the false negative error is O(e_k/phi_k). This will blow up if phi_k is small. Am I missing something?

******************** Edit: Some of my concerns about my second question is responded during the reviewer discussion period. I left the original review as it is but feel free to ignore my second question.

Summary: The authors consider the problem of active learning with access to both weak and strong labelers. It's assumed that the weak labeler provide labels from a (possibly) different joint distribution than the strong labeler. The hope is to save in the number of queries to the strong labeler compared to the plain active learning.

The authors show that this is possible under the assumption that it is possible to learn a difference classifier with small false negative error rate.

Submitted by Assigned_Reviewer_3

The authors focus on active learning from multiple annotators, especially when there are strong and weak labelers. The idea is straightforward, i.e., learn the difference between the strong and weak labelers, then query the weak labeler when the difference classifier predicts agreement while query the strong labeler when the difference classifier predicts disagreement. With the assumption that there exists a good difference classifier which has very low false negative error, the authors provide a sample complexity bound on the number of queries to the strong labelers.

The key issues in this paper are that whether there exists a good difference classifiers and that how to learn it. The proposed idea is querying both strong and weak labelers for labels and using these labels to learn the difference classifiers. How many labels are sufficient for it? Without strong assumptions, I think that the number of labels needed for this may be very close to that for learning the target concept. The authors assume that there exists a good difference classifier which has very low false negative error. Does this assumption hold? What kind of strong and weak labelers could satisfy this assumption? It is the most important issue the authors should discussion. In previous study, the exploited assumption is that the weak labeler does not make mistakes close to the decision boundary. Obviously, if the weak labeler does not make mistakes close to the decision boundary, we can learn a good difference classifier. The authors should make more efforts on how to judge whether there exists a good difference classifier between the strong and weak labelers.

The sample complexity is only analyzed for queries to the strong labeler. Does it mean that we do not care the number of queries to the weak labelers? If so, how about crowdsourcing with many weak labelers?

This paper focuses on the understanding of active learning from multiple annotatorsHowever, the authors only study a specific case of learning from strong and weak labelers with strong assumption. More efforts and discussions should be provided.
Summary: In this paper, the authors try to make a step towards the theoretical understanding of active learning from multiple annotators, e.g., what kind of assumptions are needed for active learning from multiple annotators.

Submitted by Assigned_Reviewer_4

Even though I was assigned a light review, I have read the paper in some detail, and would like to add some comments (originally written in discussion with other reviewers, slightly edited). I remain in favor of accepting the paper.

---

Two concerns about the paper were the presentation and the conditions that lead to improvements over single-oracle active learning.

Regarding presentation: I agree that the writing could be improved, but I think they do a reasonable job giving the main idea behind the algorithm and the theoretical guarantees. Indeed, there is not much attempt to give proof ideas in the paper, but I think this is because they mostly build on existing analyses of "DBAL" algorithms. The "whole subsection of the proposed algorithm" that is deferred to the appendix is not a major contribution in this paper, as it comes from previous "DBAL" algorithms (see references [4,13]), so I am comfortable with that deferral.

Regarding conditions: I agree that more discussion of the conditions would be useful, but the authors do at least provide one "case study" example in Section 4.1. Furthermore, the "assumption" is really just one that is always satisfied with large enough \eta and \alpha, but the improvement over single-oracle active learning may be minimal or non-existent when \eta and \alpha are large. So I don't think Assumption 1 (either part) is restrictive.

The way I prefer to think of the authors' setting is to pretend the weak oracle is actually a classifier that you already have in hand before learning begins. For instance, it could come from learning on different data where labels are more abundant, or it could be a preexisting classifier from a legacy system. Then clearly if you have this classifier in hand, you can make "oracle calls" very cheaply without any human effort. The question is whether this is good for anything. This paper provides one characterization of when it is useful: it is useful when it is easy to "amend" the classifier by learning the "difference" between it and the correct labeling. I think this is very cool.

So, I think in terms of both scientific merits and presentation, the paper is above threshold for acceptance.
Summary: The paper develops an interesting algorithm for active learning by combining some old techniques with some new tricks. Although I'm not sure the "weak and strong labelers" setting is the best way to position this work, I think it is a valuable contribution deserving of publication.

Author Feedback
Author rebuttal: We thank all reviewers for their support and feedback.

***Reviewer 3***

R3 appears to like the paper, but argues for rejection because most of the technical content is in the appendix, and there are many typos and formatting errors. The technical content was put in the Appendix due to space constraints. In our experience this is very standard for (the more technically involved) NIPS papers.

We can correct the formatting violations and typos quite easily, and we will follow the reviewer's suggestions to improve the notation in the final version of the paper.

***Reviewer 2***

The main contribution of this paper is a theoretical analysis of active learning with two annotators when a "gold standard" is available but expensive. Even this case was not understood theoretically before our work. We leave the general case of learning from multiple annotators for future work.

"Without strong assumptions, I think that the number of labels needed for this may be very close to that for learning the target concept."

The number of labels needed to fit the difference classifier is a **lower order term** compared to the #labels needed for active learning in the **agnostic case**. This is discussed in detail in Section 4.1.

Specifically, the cost of fitting the difference classifier is O(\theta(2\nu+\epsilon) d'(\nu+\epsilon)/\epsilon). This is of lower order than O(\theta(2\nu+\epsilon) d(\nu+\epsilon)^2/\epsilon^2) in DBAL, when d' = O(d). This is because fitting a **cost-sensitive** classifier with false negative error at most \epsilon requires less labels (O(d'/\epsilon), as opposed to O(d'/\epsilon^2)) than standard supervised learning.)

"The authors assume that there exists a good difference classifier which has very low false negative error. Does this assumption hold?"

Assumption 1 can be trivially satisfied if the constant classifier that always predicts 1 is within the class of difference classifiers. In Section 4.1, we present a concrete example where Assumption 1 holds in nontrivial cases, and our algorithm results in better label complexity than DBAL.

"The sample complexity is only analyzed for queries to the strong labeler. Does it mean that we do not care the number of queries to the weak labelers?"

The number of weak labels required by our algorithm can be trivially shown to be of the same order as standard DBAL (plus the overhead for fitting the difference classifier). This was omitted from the theorem statements for clarity of presentation.

***Reviewer 1***

"To take a step back, what is known about this problem in the passive learning other than the work of Urner, Ben-David and Shamir you cited?"

We are not aware of any other theoretical work on this topic.

"but how easy it is to learn the difference classifier? More concretely what is the relationship between d' and d? As authors mentioned, d' can be greater than d but is there an upper bound on d' (in terms of d?)"

The purpose of this paper is to provide a **general framework** with minimally restrictive assumptions to show that active learning with weak and strong labeler can be favorable than simply using the strong labeler. Therefore, H^{df} can be chosen arbitrarily, possibly making d' significantly greater than d.

"Assumption 1 seems very strong, btw, specially (3)."

(2),(3) together give a general characterization on the discrepancy between W and O.
In the worst case, \alpha(r,\eta) is at most P(DIS(B(h^*,r))), in which case we don't get an advantage over plain active learning. In order to make \alpha(r,\eta) small, two implicit assumptions have to be made: 1. W's deviation from O is small; 2. there is a difference classifier with low complexity that "fits" the difference between W and O well.

Of course, the "no free lunch" principle tells us that we cannot achieve better label complexity without assumptions on W and O.